# FAST AND PRECISE: ADJUSTING PLANNING HORIZON WITH ADAPTIVE SUBGOAL SEARCH

**Michał Zawalski** *
University of Warsaw
m.zawalski@uw.edu.pl

**Michał Tyrolski** *
University of Warsaw
michal.tyrolski@
gmail.com

**Konrad Czechowski** *
University of Warsaw
k.czechowski@
mimuw.edu.pl

**Tomasz Odrzygóźdź**
IDEAS NCBR
tomaszo@impan.pl

**Damian Stachura**
Jagiellonian University
damian.stachura1@
gmail.com

**Piotr Piękos**
KAUST[†]
piotrpiekos@gmail.com

**Yuhuai Wu**
Google Research
& Stanford University
yuhuai@google.com

**Łukasz Kuciński**
Polish Academy of Sciences
lkucinski@impan.pl

**Piotr Miłoś**
Ideas NCBR,
Polish Academy of Sciences,
deepsense.ai
pmilos@impan.pl

## ABSTRACT

Complex reasoning problems contain states that vary in the computational cost required to determine the right action plan. To take advantage of this property, we propose Adaptive Subgoal Search (AdaSubS), a search method that adaptively adjusts the planning horizon. To this end, AdaSubS generates diverse sets of subgoals at different distances. A verification mechanism is employed to filter out unreachable subgoals swiftly, making it possible to focus on feasible further subgoals. In this way, AdaSubS benefits from the efficiency of planning with longer-term subgoals and the fine control with shorter-term ones, and thus scales well to difficult planning problems. We show that AdaSubS significantly surpasses hierarchical planning algorithms on three complex reasoning tasks: Sokoban, the Rubik's Cube, and the inequality-proving benchmark INT.

## 1 INTRODUCTION

When solving hard problems, people often try to decompose them into smaller parts that are typically easier to complete (Hollerman et al., 2000). Similarly, *subgoal search methods* aim to solve complex tasks by considering intermediate subgoals leading towards the main goal. Besides their intuitive appeal, such approaches offer many practical advantages. Most notably, they enable deeper search within a smaller computational budget and reduce the negative impact of approximation errors. *Subgoal search* methods powered by deep learning have shown promising results for continuous control tasks, such as robotic arm manipulation (Nair & Finn, 2020; Jayaraman et al., 2019; Fang et al., 2019) and navigation (Kim et al., 2019; Savinov et al., 2018). Recently, Czechowski et al. (2021) showed that the usage of a subgoal generator can significantly improve search efficiency on discrete domains with high combinatorial complexity.

This paper uses Czechowski et al. (2021) as a starting point and pushes forward, building upon the following observation: many complex reasoning problems contain states that vary in complexity, measured by the computational cost required to determine the right action plan. To illustrate this, imagine driving a car. When traversing a narrow, winding street, it is crucial to focus on the closest events: the next turn, the next car to avoid, etc. However, after entering a straight, empty street, it

---

*equal contribution; Published as a conference paper at ICLR 2023, notable-top-5%.
†Work done while at the University of Warsaw

is enough to think about reaching its far end. This suggests that careful balancing of the subgoal distance is desirable: this involves selecting longer-term subgoals, if possible, to advance faster towards the goal, and choosing shorter-term subgoals to power through the harder states. Hence, the question arises whether it is possible and, if so, how to incorporate this adaptive subgoal generation procedure into subgoal search methods. In this paper, we answer this question affirmatively.

We propose a novel planning algorithm *Adaptive Subgoal Search* (AdaSubS), which adaptively chooses from subgoals with different horizons. Our method benefits both from the efficiency of planning with longer-term subgoals and from the reliability of shorter-term ones. AdaSubS *prioritizes further distances*, retracting to shorter ranges only when stuck. Additionally, we introduce a *verifier network*, which assesses whether the proposed subgoal is valid and reachable. The verifier makes it possible to efficiently discard faulty subgoals, which are common

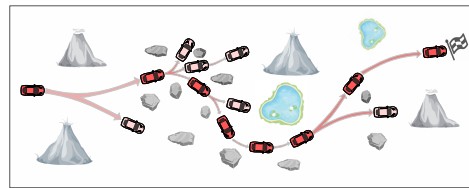

An illustrative example of adaptive planning. The planner may choose long-distance subgoals in the easier areas (e.g. the left most part) and use short distances in the hard areas (e.g. middle part).

and more costly to detect in longer horizons. AdaSubS is a data-driven algorithm whose key components are implemented as learnable deep models. In most cases, we use general-purpose transformer architectures to model subgoal generators and the verifier networks. We train those models on offline data.

We show the effectiveness of AdaSubS in three challenging domains: Sokoban, Rubik's Cube, and the inequality theorem prover INT (Wu et al., 2021). AdaSubS significantly surpasses hierarchical planning algorithms and sets a new state-of-the-art on INT.

Our main contributions are:

1. We propose Adaptive Subgoal Search (AdaSubS), a new algorithm that adjusts the planning horizon to take into account the varying complexity of the state space.

2. We present a comprehensive study of adaptive methods, showing that they outperform similar algorithms without adaptation. Amongst these, AdaSubS is the best choice across environments and planning budgets.

3. We also observe a strong indication of out-of-distribution generalization. AdaSubS trained on the proof of length 15 in INT (longest considered in the literature so far) retains more than 50% of its performance when the proof length is increased two-fold.

The code of our method is available at https://github.com/AdaptiveSubgoalSearch/adaptive_subs.

## 2 RELATED WORK

The combination of planning algorithms with deep learning is an active area of research. It provided impressive results e.g., in automated theorem proving (Polu & Sutskever, 2020), chess and Go (Silver et al., 2017), Atari benchmark (Schrittwieser et al., 2019), and video compression (Mandhane et al., 2022).

In the field of hierarchical planning, the majority of deep-learning-based methods have focused on visual domains (Kim et al., 2019; Pertsch et al., 2020a; Jayaraman et al., 2019; Fang et al., 2019) or on landmark-based navigation methods (Liu et al., 2020a; Gao et al., 2017; Zhang et al., 2020) . This body of work often relies on variational autoencoders for the compression of visual observations and uses planning mechanisms suitable for continuous control settings.

There exist many approaches to hierarchical planning utilizing different temporal distances. Kim et al. (2019) and Pertsch et al. (2020b) use hierarchical variational models to learn the temporal structure of tasks by reconstructing the visual state sequences. Pertsch et al. (2020a); Parascandolo et al. (2020); Jurgenson et al. (2020) recursively construct a plan by generating subgoals in the middle between the existing ones. Allen et al. (2021) generate macro-actions that help to speed-up the search. This differs from our work, as we use learning to generate subgoals (as opposed to action sequences) and the process is agnostic with respect to the size of the action space. These works have been shown to work on domains with limited combinatorial complexity.

Recently, Czechowski et al. (2021) has shown how combinatorially complex domains can be treated with a hierarchical planning method. Their approach shares similarities with our Adaptive Subgoal Search; however, it cannot address variable environment complexity. By using the mechanism of adaptive selection of the subgoal generation distance and verifier, we successfully tackle this problem, confirmed by significant performance gains. Our verifier is based on ideas similar to Cobbe et al. (2021); Kurutach et al. (2018); Ahn et al. (2022).

Our approach relates to established search algorithms (Cormen et al., 2009; Russell & Norvig, 2010), such as Best First Search or A*. Adaptivity techniques in the classical setup are discussed in Fickert (2022). Koenig & Likhachev (2006) propose an adaptive mechanism to improve A* by updating the goal-distance heuristic with local search. AdaSubS instead uses a fixed heuristic and adapts to the local complexity by alternating between subgoal distances. Domain-independent PDDL-based planners (McDermott et al., 1998) do not use training and attempt to solve problems in a zero-shot manner. Thus, they indicate a lower bound on performance. On the other hand, there are domain-specific methods (Korf, 1997; Büchner et al., 2022; Muppasani et al., 2022). Due to their focus, they indicate an upper bound.

AdaSubS relates to multi-queue methods (see Richter & Westphal (2010); Helmert (2006)), which alternate between multiple heuristics. Some of our planners, e.g., IterativeMixing or Longest-first, can be viewed through the lens of this approach in the sense that we could keep the priority queues for each generator separate (but with the same heuristic being a value function) and have an alternation mechanism between them. The key difference lies in the expansion phase: we expand subgoals instead of children and only the generator associated with the currently selected queue is used. [1]

Different instances of Sokoban, Rubik's Cube, and INT can be viewed as tasks with varying degrees of difficulty. Consequently, AdaSubS benefits from sharing data between these instances in a manner typical to multitask learning (Caruana, 1998). In particular, we use a goal-conditioned policy which is trained similarly as in Andrychowicz et al. (2020) or Kaelbling (1993). Additionally, the out-of-distribution generalization of AdaSubS hints at strong meta-learning capabilities of the method (Yu et al., 2020; Duan et al., 2016; Wang et al., 2016; Hessel et al., 2019).

## 3 Method

For this work, we propose Adaptive Subgoal Search (AdaSubS), a subgoal-based search algorithm designed to solve tasks that can be formulated as a search over a graph with a known transition model. AdaSubS is the best choice stemming from a careful study of methods based on the principle of mixing subgoal distances; see Section 4.4 for their definitions and empirical comparisons.

AdaSubS (see Algorithm 1) utilizes the following key components: *subgoal generators*, *verifier*, *conditional low-level policy* (CLLP), and *value function*. These components are implemented using trained neural networks (see Appendix B). To solve a task, AdaSubS iteratively builds a tree of subgoals reachable from the initial state until the target state is reached or the search budget is depleted. In each iteration, it chooses a node in the tree that is expanded by one of the generators. The chosen generator creates a few subgoal candidates, i.e., states expected to be a few steps closer to the target than the current node. For each of them, we use the verifier and CLLP to check whether they are valid and reachable within a few steps. For the correct subgoals, we compute their value function, place them in the search tree, and the next iteration follows (see Figure 1).

AdaSubS follows the general structure of Best-First Search (BestFS); thus, the key design decision is how we prioritize nodes to expand and choose generators to produce subgoals. We defer the answer to these questions after providing details of the algorithm components (see also Appendix G and the flowchart there).

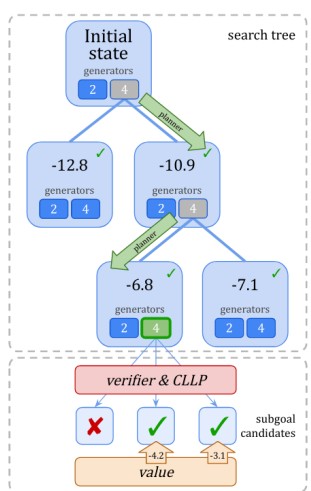

Figure 1: An example iteration of the search performed by AdaSubS.

[1]For search engines using multi-queues, see Fast downward `https://www.fast-downward.org/`; LAPKT `https://github.com/LAPKT-dev/LAPKT-public`. For PDDL generators, see `https://github.com/AI-Planning/pddl-generators`

**Subgoal generators.** The subgoal generator, or more precisely the $k$-subgoal generator, takes a state as input and returns a diverse set of new candidate states expected to be $k$ step closer to the solution. The key trade-off, which AdaSubS needs to address, is that further subgoals, i.e., those for higher values of $k$, advance faster towards the target but are also increasingly harder to generate and verify. We typically use a few (e.g., 3) generators with a list of $k$ chosen basing on experiments, namely for INT [3, 2, 1], Rubik [4, 3, 2] and Sokoban [8, 4, 2]. Note that this is the only component of AdaSubS that outputs a set of predictions.

**Conditional low-level policy (CLLP).** CLLP returns a path of low-level actions between two states (see Algorithm 2). CLLP calls iteratively a conditional low-level policy network (PN). PN takes as input the current and target states and returns an action. It is possible that CLLP is not able to reach the target, in which case an empty sequence is returned. The role of CLLP is two-fold: it serves as a mechanism allowing AdaSubS to transition between subgoals, and together with the verifier network, it is used in the subgoal verification algorithm (see Algorithm 3).

**Verifier.** The verifier network is used in the verification algorithm (see Algorithm 3), to answer the following binary classification question: given a starting state and a goal state, is it possible to reach the latter from the former using conditional low-level policy? Computationally, the evaluation of the verifier network is faster than CLLP. However, since the verifier is a binary classifier, we expect two types of error to occur: accept invalid subgoals or reject valid subgoals. The verification algorithm accepts a subgoal if the verifier network values are above a certain threshold (likewise, they are rejected if the value is below another threshold), see $\mathtt{t_{lo}}$ and $\mathtt{t_{hi}}$ in Algorithm 3. In the remaining case, the algorithm falls back on CLLP to decide whether to keep or discard a given subgoal.

**Value function.** The value function is a neural network that estimates the negative distance between the current and goal states. The planner uses this information to select the next node to expand.

**Adaptive Subgoal Search.** The particular way in which AdaSubS chooses nodes to expand and a generator to produce subgoals (see highlighted lines in Algorithm 1) implements an adaptive mechanism that adjusts the planning horizon. The key difficulty to tackle here is that further subgoals, despite being capable of advancing the search faster, are more likely to be faulty. Nevertheless, we assume an optimistic approach prioritizing the longer distances (e.g., higher $k$). If the search using the long steps gets stuck, the planner retracts and expands the most promising, high-value nodes with closer, more conservative subgoals. The verifier network helps in mitigating the risks of this strategy, as it allows for the efficient rejection of faulty subgoals. This way, by traversing easier parts using fast long-distance subgoals and conservative ones in harder parts, AdaSubS adapts to the variable complexity of the environment.

Algorithm 1 presents a simple implementation of this approach. The nodes in the search tree are placed in a max-priority queue $T$ with keys, being the pairs $(k, v)$ of the next subgoal distance and its estimated value, sorted in lexicographical order. In this way, Algorithm 1 uses the highest $k$ possible, searching with the BestFS strategy over values. If for a given $k$, all generated subgoals are invalid (faulty or unreachable), Algorithm 1 will expand for shorter distances. If successful, we go back to generating with the highest value of $k$. After reaching the target states, AdaSubS reconstructs the path of subgoals and fills it with low-level actions; see function LL_PATH in Algorithm 4.

With a slight modification, AdaSubS can be guaranteed to find a solution to any given problem, provided there is a large enough computational budget. See Appendix G.1 for details.

## 3.1 TRAINING OBJECTIVES

The components of AdaSubS are trained using a dataset of offline trajectories of subsequent states and actions: $(s_0, a_0), \ldots, (s_{n-1}, a_{n-1}), s_n$. We do not assume that they are perfect; for some of our environments, even randomly generated trajectories may turn out to be sufficient. Details on how the data is collected for each domain can be found in Section 4.1 and Appendix B.

Provided with such data, we train the $k$-generators to map $s_i$ onto $s_{i+k}$. The value function is trained to map $s_i$ onto $(i - n)$. CLLP is trained to map $(s_i, s_{i+d})$ onto $a_i$ for every $d \leq k_{\max}$ ($k_{\max}$ is the maximal distance of the subgoal generators used).

AdaSubS still works, albeit much worse if we disable the verifier network (e.g., by setting $\mathtt{t_{hi}} = 1$ and $\mathtt{t_{lo}} = 0$ in Algorithm 3). However, it is a useful setup to gather a dataset of subgoals and their reachability verified by CLLP. This dataset is used to train the verifier network, see Appendix D.

**Algorithm 1** Adaptive Subgoal Search

**Requires:**
| | |
|---|---|
| $C_1$ | max number of nodes |
| $V$ | value function network |
| $\rho_{k_0}, \ldots, \rho_{k_m}$ | subgoal generators |
| SOLVED | predicate of solution |

**function** SOLVE($\mathbf{s}_0$)
    T $\leftarrow \emptyset$    ▷ priority queue with lexicographic order
    parents $\leftarrow \{\}$
    **for** $k$ in $k_0, \ldots, k_m$ **do**
        T.PUSH($((k, V(\mathbf{s}_0)), \mathbf{s}_0)$)
    seen.ADD($\mathbf{s}_0$)    ▷ seen is a set
    **while** $0 < $ LEN(T) **and** LEN(seen) $< C_1$ **do**
        $(k, \_), \mathbf{s} \leftarrow$ T.EXTRACT_MAX()
        subgoals $\leftarrow \rho_k(\mathbf{s})$
        **for** $\mathbf{s}'$ **in** subgoals **do**
            **if** $\mathbf{s}'$ **in** seen **then** continue
            **if not** IS_VALID($\mathbf{s}, \mathbf{s}'$) **then**
                continue
            seen.ADD($\mathbf{s}'$)
            parents[$\mathbf{s}'$] $\leftarrow s$
            **for** $k$ in $k_0, \ldots, k_m$ **do**
                T.PUSH($((k, V(\mathbf{s}')), \mathbf{s}')$)
            **if** SOLVED($\mathbf{s}'$) **then**
                **return** LL_PATH($s'$, parents)
                      ▷ get low-level path, see Alg. 4
    **return** False

**Algorithm 2** Conditional low-level policy

**Requires:**
| | |
|---|---|
| $C_2$ | steps limit |
| $\pi$ | conditional low-level policy network |
| $M$ | model of the environment |

**function** GET_PATH($\mathbf{s}_0$, subgoal)
    step $\leftarrow 0$, $\mathbf{s} \leftarrow \mathbf{s}_0$
    action_path $\leftarrow []$
    **while** step $< C_2$ **do**
        action $\leftarrow \pi$.PREDICT($\mathbf{s}$, subgoal)
        action_path.APPEND(action)
        $\mathbf{s} \leftarrow M$.NEXT_STATE($\mathbf{s}$, action)
        **if** $\mathbf{s} = $ subgoal **then**
            **return** action_path    ▷ success
        step $\leftarrow$ step $+ 1$
    **return** []    ▷ subgoal is unreachable

**Algorithm 3** Verification algorithm

**Requires:**
| | |
|---|---|
| $v$ | verifier network |
| $\mathbf{t}_{\text{hi}}$ | upper threshold |
| $\mathbf{t}_{\text{lo}}$ | lower threshold |

**function** IS_VALID($\mathbf{s}, \mathbf{s}'$)
    **if** $v(\mathbf{s}, \mathbf{s}') > \mathbf{t}_{\text{hi}}$ **then return** True
    **else if** $v(\mathbf{s}, \mathbf{s}') < \mathbf{t}_{\text{lo}}$ **then return** False
    **return** GET_PATH($\mathbf{s}, \mathbf{s}'$) $\neq []$

For INT and Rubik's Cube, we use transformer models for all the key components. For the Sokoban, we utilize convolutional networks, for details see Appendix B.

## 4 EXPERIMENTS

We empirically demonstrate the efficiency of Adaptive Subgoal Search on three complex reasoning domains: Sokoban, Rubik's Cube, and the inequality proving benchmark INT (Wu et al., 2021). We demonstrate that AdaSubS is the best choice in a family of adaptive methods. Interestingly, even weaker methods in this class also outperform non-adaptive baselines. Finally, we show that AdaSubS has strong out-of-distribution generalization properties on INT.

As the performance metric, we use the success rate, defined as the fraction of solved problem instances. The computational budget is defined as the graph size, i.e., the number of nodes visited during the search and evaluated with a neural network (subgoal generator, value function, verifier, or conditional low-level policy). In Appendix C we provide details concerning the number of neural network calls, wall-time evaluations and memory usage.

### 4.1 EXPERIMENTAL DOMAINS AND DATASETS

**Sokoban** is a puzzle in which the goal is to push boxes on target locations. It is a popular testing ground for classical planning methods (Lipovetzky & Geffner, 2012), and deep-learning approaches (Guez et al., 2019; Miłoś et al., 2019). Sokoban is considered to be hard (Fern et al., 2011) due to its combinatorial complexity. Finding a solution for a given Sokoban board is an NP-hard problem. In our experiments we used $12 \times 12$ Sokoban boards with four boxes.

**Rubik's Cube** is a famous 3D puzzle with over $4.3 \times 10^{19}$ possible configurations (Korf, 1997). Recently Agostinelli et al. (2019); Czechowski et al. (2021) have developed methods for solving Rubik's Cube using neural networks.

**INT** is a benchmark for automated theorem proving proposed by (Wu et al., 2021). It consists of a generator of mathematical inequalities and a tool for proof verification. An action (proof step) in INT is a string containing an axiom and a specification of its input entities, making the action space effectively infinite and thus challenging search algorithms.

To collect offline trajectories datasets for Rubik's Cube, we generate random paths of length 20 starting from the solved cube and take them in reversed order. For INT we use the generator provided by Wu et al. (2021). For Sokoban, we use the expert data generated by a reinforcement learning agent (Miłoś et al., 2019). Detailed information is contained in Appendix D.

### 4.2 PROTOCOL AND BASELINES

Our protocol consists of three stages. In the first one, an offline dataset is prepared; see Section 4.1 and Appendix D. Secondly, we use this dataset to train the learnable components of AdaSubS: the family of subgoal generators, verifier network, and value network, see Section 3.1. Evaluation is the final step in which the algorithm's performance is verified. We measure its *success rate* using 1000 instances of a problem for each domain.

As baselines, we use BestFS and kSubS, both with the same models' checkpoints as AdaSubS. The former is a well-known class of search algorithms (including $A^*$), which, among others, performs strongly on problems with high combinatorial complexity (Pearl, 1984), achieves state-of-the-art results in theorem proving (Polu & Sutskever, 2020), and strong results on Rubik's Cube (Agostinelli et al., 2019; Czechowski et al., 2021). BestFS baseline selects actions with a trained policy network.

kSubS is the first general learned hierarchical planning algorithm proven to work on complex reasoning domains (Czechowski et al., 2021) (called BF-kSubS there), attaining good results on Sokoban and Rubik's Cube, and INT. kSubS can be view as a non-adaptive version of AdaSubS realized by a suitable hyperparameters choice: a single subgoal generator and inactive verifier ($t_{lo} = 0$, $t_{hi} = 1$).

For details on the hyperparameter choice for our method and the baselines, see Appendix F. For a more detailed description of the baselines, see Appendix E.

### 4.3 MAIN RESULTS: IN- AND OUT- OF DISTRIBUTION PERFORMANCE

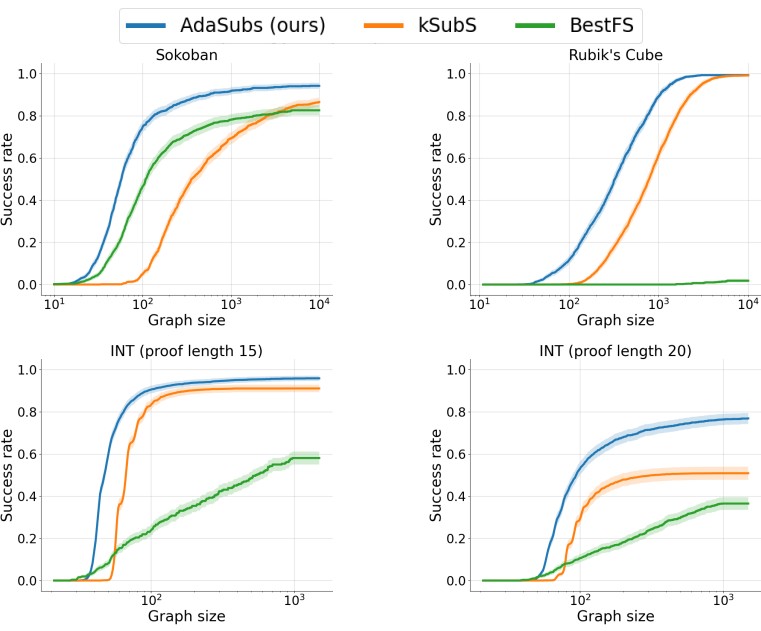

Figure 2: Success rates of AdaSubS, kSubS, and BestFS expressed in terms of graph size. The figure in the bottom right shows the out-of-distribution performance of methods evaluated on INT with proof length 20 but trained on length 15. The remaining figures present in-distribution performance. The results were measured on a fixed set of 1000 problems for each domain. Shaded areas indicate 95% confidence intervals.

AdaSubS shows strong in- and out- of distribution performance. The results for the former regime are presented in Figure 2, which shows that AdaSubS is able to make use of search capacity in the most effective manner, practically dominating other methods across the graph size spectrum. Taking a closer look at the low computational budgets, one can observe that AdaSubS achieves significantly

positive success rates while the competing methods struggle. Perhaps the most striking difference is observed for INT, where at the budget of 50 nodes AdaSubS achieves around 60% success rate, while kSubS has a success rate close to zero and BestFS does not exceed 10%. This is particularly impressive since the budget of 50 nodes is only slightly larger than three times the considered proof length. To summarize, AdaSubS performs well in low computational regimes, which can be helpful in systems that need to solve search problems under compute or memory constraints.

At the far end of the computational budget spectrum, AdaSubS still performs the best, achieving above 90% performance in each environment ($\sim$ 95% for INT, 100% for Rubik's Cube, and 93% for Sokoban). Importantly, when success rates are high, and consequently the absolute differences between methods' results seem to be low, it is instructive to think about failure rates. For instance, in the case of INT (the proof length 15), the failure rate of kSubS is 9%, almost twice the failure rate of AdaSubS. For more results on low and high budgets, see Tables 10-12 in Appendix H.4.

For the out-of-distribution analysis, we used INT, an environment designed to study this phenomenon. We investigate how methods trained on the proofs of length 15 perform on problems with longer proofs (see Figure 3). The length 15 is the longest considered in the literature (Czechowski et al., 2021). However, we go much further, studying proof lengths up to 28. AdaSubS retains more than 50% of its performance, suffering a relatively slow decay of 3.5% (on average) per one step of the proof. This stands in stark contrast to kSubS, which already loses half of its performance at length 21. AdaSubS not only outperforms kSubS at each difficulty level but also achieves the most significant advantage in the hardest problems. Additionally, we provide a full profile of the success rate with respect to the graph size for proof length 20; see the bottom right-hand corner of Figure 2. AdaSubS performs much better than the baselines, with the biggest advantage for large budgets. Results for the other environments, namely Sokoban and Rubik, are included in Appendix K.

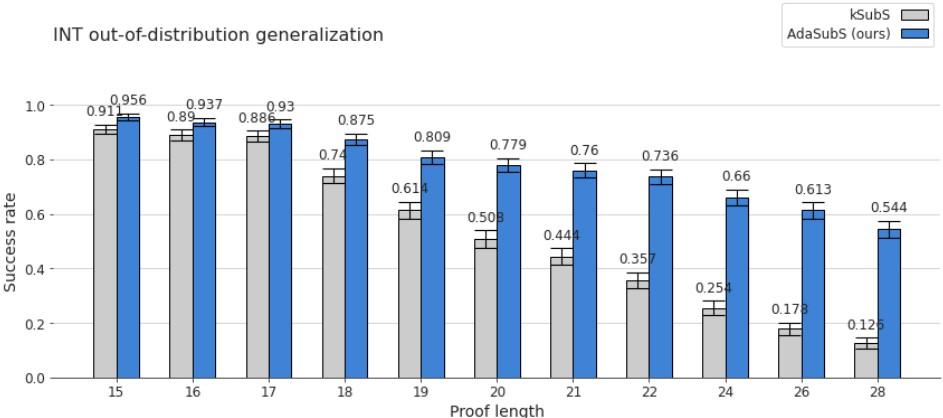

Figure 3: Out-of-distribution performance of AdaSubS and kSubS for long proofs in INT with budget of 5000 nodes. Both methods were trained on proofs of length 15. Error bars correspond to 95% confidence intervals.

The performance boost of AdaSubS over the baselines stems from two components: the verifier and the adaptativeness of subgoal selection. The former makes it possible to assign a bigger fraction of a computational budget on search by recovering a part of it from CLLP. This, in principle, could already provide significant gain when using the method. However, as shown in Table 1 and Tables 10-12 in Appendix H.4, the verifier helps, but only to a limited degree. Consequently, the majority of the improvement stems from the algorithmic novelty offered by the adaptive procedure. The adaptivity mechanism in AdaSubS creates this interesting dynamic that incentives the method to be optimistic about choosing subgoal distances while providing a safety net in case this optimism fails. How it works in practice can clearly be seen in Sokoban, where AdaSubS uses 8-subgoals 91.8% of the time, 4-subgoals 7.4% of the time, and 2-subgoals the remaining 0.8% of the time[2].

As a final note, Figure 2 can be used to infer the computational budget required for achieving a certain success rate. Additionally, the ratio of success rate to graph size can measure the efficiency

---

[2]Additionally, the 8-generator, the 4-generator, and the 2-generator generate subgoals that are on average 6.9, 3.9, and 2.0 steps away, respectively. For AdaSubS parameters, see Table 7 in Appendix F.

of the chosen budget, while the derivative of the success rate with respect to graph size provides the marginal utility of the increase in the budget.

## 4.4 Developing adaptive search methods

In this section, we present a comprehensive empirical study of four adaptive methods. This enables us to draw two main conclusions: adaptive search methods outperform non-adaptive ones, and *Longest-first* (on which AdaSubS is based) is the most efficient adaptive procedure. We present full results for INT, the hardest environment, and shortened results for Rubik and Sokoban, see Table 1. The complete set of numerical results with extended discussion can be found in Appendix H.

The four adaptive methods presented in this section are implemented using the search method[3]. Their adaptivity mechanism is defined by setting the way the subgoals are generated and the order in which the states are processed[4]. This happens in two distinguished lines in Algorithm 1 and changing them determines how the search prioritizes various distances.

| | | INT | | | |
|---|---|---|---|---|---|
| | | Small budget (50 nodes) | | Large budget (1000 nodes) | |
| | | with verifier | without | with verifier | without |
| BestFS | | - | 1.7% | - | 36.7% |
| kSubS | $k = 4$ | 2.2% | 0.1% | 82.4% | 83.0% |
| | $k = 3$ | 4.0% | 0.2% | 89.6% | 90.7% |
| | $k = 2$ | 2.1% | 0.5% | 89.8% | 91.7% |
| | $k = 1$ | 0.0% | 0.0% | 34.7% | 46.0% |
| MixSubS | $k = [4, 3, 2]$ | 0.0% | 0.0% | 94.6% | 95.0% |
| | $k = [3, 2, 1]$ | 0.0% | 0.0% | 92.2% | 92.9% |
| | $k = [3, 2]$ | 17.0% | 14.8% | 92.2% | 93.5% |
| Iterative mixing | iterations $= [1, 1, 1]$ | 32.0% | 30.1% | 87.0% | 88.6% |
| | iterations $= [10, 1, 1]$ | 43.0% | 44.8% | 95.1% | 96.0% |
| | iterations $= [4, 2, 1]$ | 54.0% | 52.1% | 93.6% | 95.5% |
| Strongest-first | | 39.5% | 40.8% | 88.5% | 89.8% |
| Longest-first | | 59.0% | 51.5% | 95.7% | 95.5% |

| Rubik (with verifier) | | |
|---|---|---|
| | 400 nodes | 6000 nodes |
| BestFS | 0.0% | 1.8% |
| kSubS | 28.8% | 98.6% |
| MixSubS | 49.1% | 99.2% |
| Iterative mixing | 50.6% | 99.1% |
| Strongest-first | 33.4% | 99.0% |
| Longest-first | 58.0% | 99.2% |

| Sokoban (small budget, 100 nodes) | | |
|---|---|---|
| | with verifier | without |
| BestFS | - | 45.9% |
| kSubS | 26.0% | 4.7% |
| MixSubS | 52.7% | 37.7% |
| Iterative mixing | 64.5% | 52.6% |
| Strongest-first | 54.6% | 41.9% |
| Longest-first | 72.2% | 63.4% |

Table 1: *(left)* Results for the INT. For each case, unless stated otherwise, the distances of subgoal generators are $k = [3, 2, 1]$. *(right)* Shortened results for Rubik and Sokoban, for complete results see Table 11 and Table 12. The results were obtained on 1000 problems each, which yields $\pm 3\%$ Bernoulli 95% confidence intervals.

Specifically, we designed and tested the following methods: *MixSubS*, *Iterative mixing*, *Strongest-first*, *Longest-first*. Each method uses a set of $n$ generators $\rho_{k_1}, \ldots, \rho_{k_n}$ trained to produce subgoals on different distances $k_1 < \ldots < k_n$ (recall Section 3.1 for training details). A more detailed description of the methods (and pseudocodes) can be found in Appendix H.

- *MixSubS* is the simplest approach, in which for each processed state we generate one subgoal from each generator $\rho_{k_i}$ (subgoals $\leftarrow \cup_{j=1}^n \rho_{k_j}(\texttt{s})$). In each iteration, *MixSubS* chooses a state with the highest value estimation $V(s)$ to process.

- *Iterative Mixing* is similar to *MixSubS* and enables for advanced schedules of generators to be used. In the consecutive iterations, the $i$-th generator is used to expand $l_i$ nodes before switching to the next generator. This allows us to flexibly prioritize the better generators, but at the cost of tuning additional hyperparameters $l_1, \ldots, l_n$. For these reasons, it is not practical, but useful as a reference point.

- *Strongest-first* uses one generator at a time (subgoals $\leftarrow \rho_{k_\ell}(\texttt{s})$), where $k_\ell$ is the longest distance not previously used in s. In each iteration, *Strongest-first* chooses a state with the highest value estimation $V(s)$ to process.

- *Longest-first* prioritizes long subgoals over the whole search procedure. Only if the queue does not contain any nodes with the highest $k$, it uses subgoals of lower distances. The nodes are processed in the order of their value estimation $V(s)$.

---

[3] Adaptivity may also be implemented using the subgoal generator. We considered various approaches in this category, however, they did not perform better than the non-adaptive baseline kSubS, see Appendix H.3. We speculate that assessing the state difficulty is a hard learning problem that is easier to handle via search.

[4] For similar considerations in classical planning, see multi-heuristic best-first search (Helmert, 2006).

The high-level empirical conclusion is that the performance of methods is roughly ordered as follows: *Longest-first > Iterative mixing > MixSubS > Strongest-first > kSubS > BestFS*.

In more detail, already the simple *MixSubS* works better than the non-adaptive baselines. In particular, it can outperform the maximum of performances of kSubS for each $k$. This is in line with the intuition that our mixing mechanism can elicit benefits of various distances while avoiding their drawbacks. We conjecture that whenever a single generator begins to struggle, the search advances with the help of another generator, allowing for stable progress. *Iterative mixing* is able to exhibit strong performance; however, it needs tedious schedule tuning for each domain.

*Strongest-first* and *Longest-first* implement bias towards longer distances. Even though they are quite similar, they display a large performance difference. We speculate that when *Strongest-first* encounters an area with falsely large value estimates, it wastes a lot of compute to examine it with all subgoal distances. On the other hand, *Longest-first* first explores other areas before using shorter subgoals and thus is able to avoid this problem. We stress that these effects are far from being obvious; however, they occur robustly across our test scenarios.

The verifier is beneficial on small budgets, especially when long subgoals are used. For large budgets, gains diminish. However, a properly tuned verifier never decreases the results significantly.

## 5 LIMITATIONS AND FUTURE WORK

**Determinism, access to model dynamics** Our main focus is combinatorially complex domains. There are many applications of interest in which we can assume access to underlying dynamics and determinism (for example Automated Theorem Proving). Nevertheless, it is an interesting future direction to adjust our method to stochastic domains and learned models.

**Reliance on the offline data** In our experiments, we need offline datasets of successful trajectories. We leave for future work developing an algorithm based on the expert iteration paradigm.

**Path optimization** The goal of our algorithm is to find any path leading to the solution. In many real-world problems, it is also important to find a short path or one with a high reward (see Appendix I for optimality measures for Sokoban).

**Completeness and memory constraints** Our algorithm is not guaranteed to find a solution. We found it is not problematic in practice. If such a property is required, it can be assured by a simple change, see Appendix G.1. However, since AdaSubS keeps a list of visited nodes (see Algorithm 1) and caches some computations (see Algorithm 4), it requires memory proportional to the search budget, which can grow up to the size of the state space when seeking completeness.

**Adversarial configuration** The performance of AdaSubS can deteriorate when the ability to train its components is reduced and the environment is hard. For example, 'walk-the-line' environment (with states either lying on a unique solving trajectory or leading to deadstates) and no training data.

**Combine with recursive search methods** In some domains, one can generate useful subgoals for long distances and recursively split the problem (Pertsch et al., 2020a; Parascandolo et al., 2020; Jurgenson et al., 2020). It would be interesting to propose an algorithm that automatically detects when such an approach is possible and combine two ways (our and recursive) of generating subgoals.

## 6 CONCLUSIONS

We study planning methods that adapt to the local complexity of a solved problem. We concentrate on the adaptive selection of the subgoal distance realized by mixing various subgoal generators. We prove that methods based on this principle outperform non-adaptive counterparts They can tackle complex reasoning tasks as demonstrated on Sokoban, the Rubik's Cube, and INT. Our main algorithm, AdaSubS, is the best of the tested choices across all environments and search budgets. Interestingly, AdaSubS exhibits high out-of-distribution generalization capability, retaining much of its performance for instances of harder problems on INT than it was trained for.

## 7   ACKNOWLEDGMENTS AND DISCLOSURE OF FUNDING

The work of Michał Zawalski and Piotr Miłoś was supported by the Polish National Science Center grant 2021/43/O/ST6/02478. This research was supported by the PL-Grid infrastructure. Our experiments were managed using `https://neptune.ai`. We would like to thant the Neptune team for providing us access to the team version and technical support.

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

## A  LOW-LEVEL PATH FUNCTION

The low-level path function (see LL_PATH, Algorithm 4) computes a path from the starting state to the goal state in the environment using low-level actions. However, it not only responsible for returning the path but also for checking false positive errors of the verifier. Specifically, the verifier can accept an unreachable state in Algorithm 3 and then wrongly include it in the solution path. Thus, LL_PATH has to construct a low-level path and confirm that every step on the way is achievable.

---

**Algorithm 4** Low-level path

---

**function** LL_PATH($s$, parents)
    ▷ parents is the dictionary of parent nodes in the subgoal tree. (S,C) ∈ parents means that C is a subgoal for state S
    path ← []
    **while** $s$ **in** parents.KEYS() **do**
        subgoal_path ← GET_PATH(parents[$s$], $s$)        ▷ see Algorithm 2.
                ▷ In practice, to reduce the number of neural network calls, we cache
                ▷ the results of the GET_PATH calls in Alg 1 and reterive them here.
        **if** subgoal_path = [] **then return False**        ▷ mistake of the verifier
        path ← concatenate(subgoal_path, path)
        $s$ ← parents[$s$]
    **return** path

---

## B    TRAINING DETAILS

### B.1    ARCHITECTURES

**INT and Rubik's cube**. All components of AdaSubS utilize the same architecture. Specifically, we used mBart, a transformer from the HuggingFace library (see (Liu et al., 2020b)). To make training of the model and the inference faster we reduced the number of parameters: we used 45M learned parameters instead of 680M in the original implementation. We used 6 layers of encoder and 6 layers of decoder. The dimension of the model was set to 512 and the number of attention heads to 8. We adjusted the size of the inner layer of position-wise fully connected to 2048. During the inference, we used beam search with width 16 for INT and width 32 for Rubik's Cube. Our implementation of the model follows (Czechowski et al., 2021, Appendix B.1)

**Sokoban**. We used four convolutional neural networks: the subgoal generator, conditional low-level policy, value, and the verifier. They all share the same architecture with a different last layer, depending on the type of output. Each model had 7 convolutional layers with kernel size (3,3) and 64 channels. Conditional low-level policy and verifier need two Sokoban boards as an input, so for these networks we concatenate them (across the last, depth dimension) and we treat two boards as one tensor. For the value function on top of a stack of convolutional layers there is a fully connected layer with 150 outputs representing 150 distances to the goal or. CLLP has analogous final layers with the one exception that there are only two classes: determining whether it is possible to reach a subgoal by CLLP or not. The network used for generatiing subgoals returns two outputs: distribution over possible modifications of a given state, and prediction whether a modified state is a good subgoal. The first output is obtained with a fully connected layer, the second with global average pooling followed by a fully connected layer. Generation of a single subgoal is realised as a sequence of calls to this network. We start from a given state and iteratively apply modifications with high probability assigned by the first head of the network, until the second head predict that no more iterations are needed. (see also Appendix G.1)

### B.2    TRAINING PIPELINE

To ensure a fair comparison with (Czechowski et al., 2021) we followed their settings for a training pipeline.

**INT and Rubik's Cube**. To train the models we used the training pipeline from the HuggingFace library (Liu et al., 2020b). We trained our models from scratch without using any pretrained checkpoints. The size of the training batch was 32, the dropout was set to 0.1, and there was no label smoothing. We used the Adam optimizer with the following parameters: $\beta_1 = 0.9$, $\beta_2 = 0.999$, $\epsilon = 10^{-8}$. We applied the warm-up learning schedule with 4000 warm-up steps and a peak learning rate of $3 \cdot 10^{-4}$. For inference in INT, we used temperature 1 and for Rubik's Cube to 0.5 (the optimal value was chosen experimentally).

**Sokoban**. To train of all networks we used a supervised setting with learning rate $10^{-4}$ and trained for 200 epochs. We used the Adam optimizer with $\beta_1 = 0.9$, $\beta_2 = 0.999$ and $\epsilon = 10^{-7}$.

### B.3    DATASETS

For dataset used to train all the network see Appendix D.

## C    Computational budget analysis

The default metric of the graph size that we use for comparisons counts all the states visited during the search, both high-level subgoals and intermediate states passed by the CLLP. It is a good estimate of the number of steps the algorithm takes to solve the given problem. For completeness, in this section, we analyze the total number of calls to every learned component of the pipeline for AdaSubS and the baseline kSubS.

Since all of the main components are deep neural networks, their evaluation time dominates the computational budget. Tables 3, 2 and 4 present the average number of calls to each component in 1000 test episodes, fixed for all the methods. That indicates which component consumes the largest part of the computational budget. The results are presented for different numbers of beams (see Appendix G.1) used for sampling predictions from the subgoal generators, the only component that outputs a set of predictions. The default number of beams was 16 for Sokoban and INT, and 32 for the Rubik's Cube (see Appendix F for the complete list of the parameters).

As the tables show, not only does AdaSubS solve more problems within smaller search graphs but also calls each component fewer times, which results in faster inference.

In the Rubik's Cube, the calls to the generators dominate the computations. However, when using smaller beams, this number can be significantly reduced while preserving the high success rate. In all the environments, AdaSubS is less sensitive to reducing the number of beams than kSubS in terms of performance. This is the case since in AdaSubS every single generator creates fewer subgoal candidates (see Tables 7-9), and thus it does not require a wide beam search. Therefore, by reducing the number of beams, AdaSubS can provide strong results within a much shorter time.

In Rubik's Cube and Sokoban, using the verifier in AdaSubS significantly reduces the number of calls to the low-level policy. However, in INT this is not the case. In most cases when kSubS fails to find a solution, at some point it cannot create any valid subgoal, and thus the search ends early. AdaSubS does not suffer from this issue, since it uses more generators. Thus, it counts the calls even from hard instances that require much larger graphs.

As shown in Table 5, if we count the calls only for the tasks solved by both methods, AdaSubS provides an advantage. Therefore, AdaSubS indeed provides better results within a smaller computational budget compared to kSubS.

| Environment | Rubik's Cube | | | | | |
|---|---|---|---|---|---|---|
| Variant | kSubS (32 beams) | kSubS (4 beams) | AdaSubS (32 beams) | AdaSubS (8 beams) | AdaSubS (4 beams) | AdaSubS (2 beams) |
| Success rate | 98.8 | 97.1 | 99.2 | 99.2 | 99 | 98.5 |
| Generator calls | 6085 | 852 | 8872 | 2205 | 1244 | 680 |
| Verifier calls | 0 | 0 | 277 | 275 | 311 | 340 |
| Policy calls | 1330 | 1526 | 352 | 350 | 395 | 446 |
| Value calls | 259 | 285 | 163 | 162 | 181 | 197 |
| Total calls | 7675 | 2664 | 9666 | 2994 | 2133 | 1665 |
| Wall-time | 86.3 sec | 49.9 sec | 96.1 sec | 49.2 sec | 47.3 sec | 37.3 sec |

Table 2: Comparison of the average number of calls to the generator, verifier, policy, and value networks for different numbers of beams (width of beam search in subgoal generation)) and the average wall time. Results were obtained using fixed 1000 instances of Rubik's Cube

### C.1    Memory usage

AdaSubS keeps track of the search tree composed of high-level nodes. Thus, the amount of required memory grows linearly with the search budget. However, if we use longer subgoals, the tree is sparse because we do not store the nodes visited by the low-level policy.

Note that the BestFS baseline, which uses only low-level steps, usually requires much larger memory because it must record every step. In practice, when evaluating the BestFS baseline in the INT

environment, we often had problems with experiments crashing because of exceeding the memory limit on the machine. We never observed similar issues when running AdaSubS.

| Environment | Sokoban | | | | | |
|---|---|---|---|---|---|---|
| Variant | kSubS (16 beams) | kSubS (8 beams) | kSubS (4 beams) | AdaSubS (16 beams) | AdaSubS (8 beams) | AdaSubS (4 beams) |
| Success rate | 84.4 | 84.6 | 82.3 | 94 | 94 | 94.1 |
| Generator calls | 2500 | 1281 | 746 | 3389 | 1692 | 848 |
| Verifier calls | 0 | 0 | 0 | 211 | 211 | 212 |
| Policy calls | 4576 | 4693 | 5554 | 248 | 247 | 247 |
| Value calls | 183 | 187 | 216 | 82 | 82 | 82 |
| Total calls | 7260 | 6161 | 6301 | 3931 | 2233 | 1391 |
| Wall-time | 48.7 sec | 36.6 sec | 33.6 | 54.4 | 39.3 sec | 30.6 sec |

Table 3: Comparison of the average number of calls to generator, verifier, policy, and value networks for different number of beams (width of beam search in subgoal generation)) and the average wall-time. Results were obtained using fixed 1000 instances of Sokoban

| Environment | INT | | | | |
|---|---|---|---|---|---|
| Variant | kSubS (16 beams) | kSubS (4 beams) | AdaSubS (16 beams) | AdaSubS (8 beams) | AdaSubS (4 beams) |
| Success rate | 90.7 | 89.7 | 96 | 96 | 95.3 |
| Generator calls | 107 | 26.0 | 362 | 166 | 76.3 |
| Verifier calls | 0 | 0 | 67.9 | 62.2 | 57.2 |
| Policy calls | 378 | 366 | 801 | 738 | 659 |
| Value calls | 6.9 | 6.6 | 14.0 | 13.0 | 11.9 |
| Total calls | 492 | 399 | 1245 | 974 | 805 |
| Wall-time | 15.3 sec | 12.1 sec | 43.8 sec | 31.9 sec | 31.1 sec |

Table 4: Comparison of the average number of calls to generator, verifier, policy, and value networks for different number of beams (width of beam search in subgoal generation)) and the average wall-time. Results were obtained using fixed 1000 instances of INT problems.

| Environment | INT | |
|---|---|---|
| Variant | kSubS | AdaSubS |
| Generator calls | 93.4 | 102 |
| Verifier calls | 0 | 19 |
| Policy calls | 328 | 300 |
| Value calls | 6.2 | 5.6 |
| Total calls | 428 | 427 |

Table 5: Comparison of the average number of calls to generator, verifier, policy, and value networks for problems solved by both methods for INT environment.

# D    DATASETS AND DATA PROCESSING

**Sokoban**. To collect offline data for Sokoban we used an MCTS-based RL agent from (Miłoś et al., 2019). In effect, the dataset consisted of all successful trajectories obtained by the agent: $154000$ trajectories for 12x12 boards with four boxes. We use $15\%$ of states from each trajectory to create the training dataset $\mathcal{D}$. We performed the split of dataset $\mathcal{D}$ into two parts of equal size: $\mathcal{D}_1$ and $\mathcal{D}_2$. The former was used to train the subgoal generators and conditional low-level policy, while the latter was used to train the verifier network. This split mitigates the possibility of the verifier's over-optimism concerning the probability of achieving subgoals by CLLP.

**INT**. We represent both states and actions as strings. For states, we used an internal INT tool for such representation. For actions, we concatenate one token representing the axiom and the arguments for this axiom (tokenized mathematical expressions) following (Czechowski et al., 2021).

To generate the dataset of successful trajectories we used the configurable generator of inequalities from the INT benchmark (see (Wu et al., 2021)). We adjusted it to construct trajectories of length 15 with all available axioms. The dataset used for our experiments consisted of $2 \cdot 10^6$ trajectories.

**Rubik's Cube**. To construct a single successful trajectory we performed 20 random permutations on an initially solved Rubik's Cube and took the reverse of this sequence, replacing each move with its reverse. Using this procedure we collected $10^7$ trajectories. Such solutions are usually sub-optimal, since random moves are not guaranteed to increase the distance from the solution. They can even make loops in the trajectories.

## D.1    DATASET FOR VERIFIER

The verifier answers the question of whether a given subgoal is reachable by the CLLP. Thus, the dataset for training this component cannot be simply extracted from the offline trajectories.

To get the training samples for the Rubik's Cube and INT, we run AdaSubS without the verifier. In other words, we set $\mathtt{t_{hi}} = 1$ and $\mathtt{t_{lo}} = 0$, which essentially means that the reachability of all the subgoal candidates is checked solely by CLLP. During the searching, the generators create subgoal candidates, which are then verified by CLLP. Therefore, after working on some problem instances, we obtain a reach dataset of valid and not valid subgoals, marked by CLLP.

For the experiments in Sokoban, the limitation of the size of the offline dataset is an important factor for the final performance. Therefore, to ensure a fair comparison of AdaSubS with baselines, we do not generate additional solutions. Instead, we split the dataset as described above into $\mathcal{D}_1$ and $\mathcal{D}_2$ and used only $\mathcal{D}_2$ to generate data for the verifier. From every trajectory in $\mathcal{D}_2$, we sample some root states. For every such state, we use the subgoal generators to predict subgoal candidates. Then, CLLP checks the validity of each of them and we include them in the verifier training dataset.

After collection, it is essential to balance the dataset. Easy instances with short solutions provide fewer datapoints than hard tasks that require a deep search. Thus, it may happen that a substantial fraction of data collected this way comes from a single instance, reducing the diversity of the dataset. We have observed such issues, particularly in the INT environment. To prevent this, during the collection of the data for INT, we limit the datapoints that can be collected from a single problem instance to at most 100. This way, we collected about $5 \cdot 10^6$ training samples for the verifier for each domain.

# E    BASELINES

As baselines, we use BestFS and BF-kSubS.

**BestFS** is a well-known class of search algorithms (including $A^*$), which, among others, performs well on problems with high combinatorial complexity (Pearl, 1984), achieves state-of-the-art results in theorem proving (Polu & Sutskever, 2020), and strong results on Rubik's Cube (Agostinelli et al., 2019; Czechowski et al., 2021).

Similarly to AdaSubS, BestFS iteratively expands the graph of visited states by choosing nodes with the highest value and adding its children to the priority queue. However, instead of using children from the subgoal tree, it uses direct neighbors in the environment space. In other words, we use a single policy network to generate neighbor subgoals in the distance of 1 action from a given node and treat it as a new subgoal. One can implement BestFS by replacing the call to a subgoal generator $\rho_k$ in Algorithm 1 with $\rho_{BFS}$.

The implementation of $\rho_{BFS}$ differs slightly between environments. In Rubik's Cube, $\rho_{BFS}$ for each action estimates the probability that it leads to the solution. In every iteration, we take the top 3 predictions. In Sokoban, we use the same training objective, but instead of taking a fixed number of actions, we take the smallest subset of actions with estimated probabilities summing to at least 98%. For INT, we use beam search to generate high-probability actions. This is necessary since in INT the actions are represented as sequences.

---

**Algorithm 5** BestFS

**Requires:**       $V$         value function network
                $\rho_{BFS}$      policy
               SOLVED     predicate of solution

**function** SOLVE($\mathbf{s_0}$)
    T $\leftarrow \emptyset$                                                                   $\triangleright$ priority queue
    parents $\leftarrow \{\}$
    T.PUSH($(V(\mathbf{s_0}), \mathbf{s_0})$)
    seen.ADD($\mathbf{s_0}$)                                                        $\triangleright$ seen is a set
    **while** $0 < $ LEN(T) and LEN(seen) $< C_1$ **do**
        _, s $\leftarrow$ T.EXTRACT_MAX()
        actions $\leftarrow \rho_{BFS}(\mathbf{s})$
        **for** a **in** actions **do**
            $\mathbf{s'} \leftarrow$ ENV_STEP($\mathbf{s, a}$)
            **if** $\mathbf{s'}$ **in** seen **then**
                continue
            **if** not IS_VALID($\mathbf{s, s'}$) **then**
                continue
            seen.ADD($\mathbf{s'}$)
            parents[$\mathbf{s'}$] $\leftarrow s$
            T.PUSH($(V(\mathbf{s'}), \mathbf{s'})$)
            **if** SOLVED($\mathbf{s'}$) **then**
                **return** LL_PATH($s'$, parents)
                                                               $\triangleright$ get low-level path, see Alg. 4
    **return** False

---

**BF-kSubS** is the first, according to our knowledge,  general learned hierarchical planning algorithm shown to work on complex reasoning domains (Czechowski et al., 2021), attaining strong results on Sokoban and Rubik's Cube and state-of-the-art results on INT. BF-kSubS is a special case of AdaSubS with the following hyperparameters choice: a single subgoal generator and inactive verifier (with $\mathtt{t_{lo}} = 0$ and $\mathtt{t_{hi}} = 1$) in Algorithm 3.

## F  HYPERPARAMETERS

| Environment | Sokoban | Rubik's Cube | INT |
|---|---|---|---|
| learning rate | $10^{-4}$ | $3 \cdot 10^{-4}$ | $3 \cdot 10^{-4}$ |
| learning rate warmup steps | - | 4000 | 4000 |
| batch size | 32 | 32 | 32 |
| kernel size | [3, 3] | - | - |
| weight decay | $10^{-4}$ | - | - |
| dropout | - | 0.1 | 0.1 |

Table 6: Hyperparameters used for training.

| Environment | Sokoban | | |
|---|---|---|---|
| Method | kSubS | MixSubS | AdaSubS (ours) |
| number of subgoals | 4 | 1 | 1 |
| number of beams | 16 | 16 | 16 |
| beam search temperature | 1 | 1 | 1 |
| $k$-generators | 8 | [8, 4, 2] | [8, 4, 2] |
| number of steps to check ($C_2$) | 10 | [10, 6, 4] | [10, 6, 4] |
| max steps in solution check | - | 18 | 18 |
| max nodes in search tree ($C_1$) | 5000 | 5000 | 5000 |
| acceptance threshold of verifier ($t_{hi}$) | - | 0.99 | 0.99 |
| rejection threshold of verifier ($t_{lo}$) | - | 0.1 | 0.1 |

Table 7: Hyperparameters used for evaluation in the Sokoban environment.

| Environment | the Rubik's Cube | | |
|---|---|---|---|
| Method | kSubS | MixSubS | AdaSubS (ours) |
| number of subgoals | 3 | 1 | 1 |
| number of beams | 32 | 32 | 32 |
| beam search temperature | 0.5 | 0.5 | 0.5 |
| $k$-generators | 4 | [4, 3] | [4, 3, 2] |
| number of steps to check ($C_2$) | 4 | [4, 3] | [4, 3, 2] |
| max steps in solution check | - | 4 | 4 |
| max nodes in search tree ($C_1$) | 5000 | 5000 | 5000 |
| acceptance threshold of verifier ($t_{hi}$) | - | 0.995 | 0.995 |
| rejection threshold of verifier ($t_{lo}$) | - | 0.0005 | 0.0005 |

Table 8: Hyperparameters used for evaluation in the Rubik's Cube environment.

Most of the hyperparameters, both for training and evaluation, are either default or have little impact on the performance of the algorithms. The values are in line with (Czechowski et al., 2021), which ensures fair comparison.

The parameter $C_1$ (see Algorithm 1) controls the number of high-level nodes in the search tree. It is lower than the actual graph size that we use for comparisons since it counts neither the intermediate states visited by CLLP nor the subgoals that turned out to be invalid. That hyperparameter was chosen to enable all the algorithms evaluated to reach the graph size values used for comparison in Figure 2 and others in Section 4.

### F.1  TUNING THE HYPERPARAMETERS

The most important training hyperparameter that has to be tuned is the learning rate. To do this, we compared the prediction accuracy for ten training runs corresponding to ten learning rates in

| Environment | INT | | |
|---|---|---|---|
| Method | kSubS | MixSubS | AdaSubS (ours) |
| number of subgoals | 4 | 2 | 3 |
| number of beams | 16 | 16 | 16 |
| beam search temperature | 1 | 1 | 1 |
| $k$-generators | 3 | [3, 2, 1] | [3, 2, 1] |
| number of steps to check ($C_2$) | 3 | [3, 2, 1] | [3, 2, 1] |
| max steps in solution check | - | 5 | 5 |
| max nodes in search tree ($C_1$) | 400 | 400 | 400 |
| acceptance threshold of verifier ($t_{hi}$) | - | 1 | 1 |
| rejection threshold of verifier ($t_{lo}$) | - | 0.1 | 0.1 |

Table 9: Hyperparameters used for evaluation in the INT environment.

the range $[10^{-5}; 10^{-3}]$. We shared the training hyperparameters across all the components for each environment, as they share the same network architecture.

In evaluation, the most important hyperparameter of AdaSubS is the set of $k$-generators. To select the set for Sokoban, we started with small values of $k$ (e.g., 1 or 2) and increased $k$ multiplicatively, doubling it as long as the new set of generators performed better. We used a similar procedure in Rubik's Cube and INT, but due to the bounds on the optimal path length (at most 26 and 15, respectively), we increased $k$ additively (incrementing $k$ by one). This way, we have chosen $k = [8, 4, 2]$ for Sokoban, $k = [4, 3, 2]$ for Rubik's Cube, and $k = [3, 2, 1]$ for INT. When evaluating a set of generators, we also need to set the number of subgoals each generator should output. Thus, each time we tried three different values ($\{1, 2, 3\}$ for Rubik's Cube and Sokoban, $\{2, 3, 4\}$ for INT) and used the one that resulted in the highest success rate. We ran the evaluation pipeline 15 times to tune that parameter. The other parameters have rather minor impact on the results, and thus we did not tune them extensively.

For kSubS, we took the values of $k$ used in (Czechowski et al., 2021) for Rubik's Cube and INT. For Sokoban, we use $k = 8$ since it outperforms $k = 4$ proposed in this work.

In the case of the verifier thresholds, we want high precision and high recall (see discussion in Section 4.5). While "loose" thresholds increase the processing speed, "tight" thresholds usually result in higher solved rates. To solve this trade-off, we propose to set the thresholds giving roughly 99% recall for the lower threshold and 99% precision for the upper. After estimating the parameters of the verifiers, it turned out that in the case of Sokoban, Rubik's Cube, and INT, the thresholds should be in the intervals $[0; 0.1]$ and $[0.9; 1]$. The final values were determined by running a grid search over five values in each interval. That required running the evaluation about 25 times. Note that since the verifier is used to increase the processing speed, it is enough to tune the thresholds only for the final version of the pipeline.

We train INT and Rubik's Cube components on a single GPU for about three days. The components for Sokoban are trained on CPUs for about a day. The evaluations in all the environments were also performed on CPU nodes. See Appendix J for the details of the actual infrastructure used in this project. Therefore, to properly tune the parameters, requires ten training runs on GPU and 40 evaluations on CPU nodes for each environment.

# G  Components of AdaSubS

## G.1  Subgoal generators

The main purpose of the subgoal generator is to propose subgoal candidates in every iteration of the planner loop. That is, given the current state $s$ of the environment it should output other states that are a few steps closer to goal $g$.

We train a $k$-generator by extracting training data from successful trajectories. Let $s_0, s_1, \ldots, s_n$ be a trajectory that leads to the goal $s_n$. For every state $s_i$ we train the $k$-generator network to output the state $s_{i+k}$, exactly $k$ steps ahead. Provided with a dataset of trajectories, this is a supervised objective. Clearly, a state that is $k$ steps ahead does not need to be exactly $k$ steps closer to the solution, especially if the trajectories include noise or exploration. However, it is guaranteed to be at most $k$ steps closer, which enables reliable limits to be set for checking reachability.

In a simple approach, $k$ is a hyperparameter that needs to be fixed, as proposed by (Czechowski et al., 2021). However, this is a strong limitation if the environment exhibits a variable complexity problem. Therefore, AdaSubS instead uses a set of generators, trained for different values of $k$. This way, the planner can adjust the expansion to match the local complexity of the problem. Additionally, training a set of generators can be easily parallelized.

For our generators, we use the transformer architecture. The input state is encoded as a sequence of tokens, as described in (Czechowski et al., 2021, Appendix C). The network produces another sequence of tokens on the output, which is then decoded to a subgoal state. The output sequence is optimized for the highest joint probability with beam search: the consecutive tokens are sampled iteratively and a fixed number of locally best sequences passes to the next iteration. This way, the generator enables sampling of a diverse set of subgoals by adjusting the beam width and sampling temperature. The exact number of the subgoals that the generators output are given in Appendix F.

As noted in Section 3.1, for the Sokoban environment instead of transformers we use simple convolutional networks. In this domain, the subgoal is created by a sequence of changes to the input state. The generator network is trained to predict the probability of changing every pixel. Then, the subgoals are obtained as a sequence of modifications that maximize joint probability. For simplicity, in AdaSubS we use beam search for all domains, including Sokoban.

During the inference, the generators output a limited set of subgoals to explore. Thus, if the generators are not trained well enough, even with an infinite computational budget, the search may fail to find a solution to the given problem, even if one exists. However, with a slight modification, AdaSubS can be guaranteed to find a solution to any given problem (or correctly report that the solution does not exist). We achieve that by adding an exhaustive single-step policy as the last generator. It would populate an empty queue with all children of the highest valued but not yet expanded node. Note that such a modification never decreases the score since the dummy generator is only used when a search is about to fail. In practice, this type of modification is not necessary to obtain strong results.

## G.2  Conditional low-level policy (CLLP)

If we want to add a subgoal candidate to our search tree, we need to check whether it is reachable from the current state. This can be done using CLLP – a mapping that given a state and a subgoal produces a sequence of actions that connects those configurations, or claims that there is no such configuration. Specifically, the policy network, given the state and subgoal, iteratively selects the best action and executes it until the subgoal is reached or a threshold number of steps is exceeded, as shown in Algorithm 2.

CLLP is trained to imitate the policy that collected the training data. For every pair of states $s_i, s_j$ that are located at most $d$ steps from each other, it is trained to predict action $a_i$, taken in state $s_i$. Such action may not be optimal but usually it leads closer to $s_j$. The threshold $d$ controls the range of the policy, as it is trained to connect states that are at most $d$ steps away. Thus, it is essential to set the hyperparameter $d$ to a value that is greater than the distances of all the generators used.

It is essential that the policy can successfully reach the correct subgoals, as it is a necessary condition for adding them to the search tree. The training metrics show that in all our environments the policy can reach more than $95\%$ of correct subgoals. This percentage is even higher for short distances.

### G.3 VERIFIER

To check whether a $k$-subgoal is reachable with the conditional policy, we need to call it up to $k$ times. If we decide to use generators with long horizons, it becomes a significant computational cost. To mitigate this issue, we use the verifier that estimates the validity of a subgoal candidate in a single call. During the search, the generated subgoal candidates are evaluated by the verifier. For each of them, it estimates whether they are valid and outputs its confidence. If the returned confidence exceeds a fixed threshold, we do not run the costly check with the conditional policy. We perform such a check only in case the verifier is uncertain (see Algorithm 3).

At the end of the search, when a solving trajectory is found, we need to find the paths between all the pairs of consecutive subgoals that were omitted due to the verifier (see Algorithm 4). Since the length of the final trajectory is usually much smaller than the search tree, that final check requires much less computations.

It should be noted that the verifier estimates validity with respect to the conditional policy that is used. In case a valid subgoal is generated but the policy cannot reach it for some reason, it cannot be used to build the search tree anyway, for no solution that uses it can be generated in the final phase. Thus, the verifier should be trained to predict whether the CLLP that is used can reach the subgoal, rather than whether it is reachable by an optimal policy.

To train the verifier, we run our pipeline on some problem instances. All the subgoals created by the generators are validated with CLLP. This way, eventually we obtain a dataset of reachable and unreachable subgoal candidates. We train the verifier network to fit that data. Unlike the other components, training the verifier does not require access to any trajectories, only to a number of problem instances.

### G.4 VALUE FUNCTION.

The value function $V : \mathcal{S} \to \mathbb{R}$ estimates the negative distance between the current state $s$ and the goal state $g$. During the search, this information is used to select the most promising nodes to expand. For every trajectory $s_0, \ldots, s_n$ in the dataset it is trained to output $i - n$ given $s_i$. We opted for a simple training objective but any value function can be used in the algorithm.

### G.5 QUALITY OF ADASUBS COMPONENTS

The effectiveness of AdaSubS depends on the quality of its trainable components: below we present an analysis of generators and verifiers.

**Generator: $k$ trade-off**. The quality of generators deteriorates when $k$ is increased. In Sokoban, nearly $90\%$ of subgoals created with the 4-generator are valid (according to CLLP). However at the same time for the 16-generator, this figure drops to $50\%$. Similarly, in Rubik's Cube, about $82\%$ of subgoals proposed by the 4-generator are valid, while the 3-generator has over $99\%$ accuracy. On the other hand, longer distances are beneficial to the search as they make it possible to build sparser search trees and achieve a solution faster. It turns out that the optimal choice of $k$ depends on the search budget. It pays to be optimistic (i.e., choose long distances) for small budgets, while more prudent choices have the upper hand

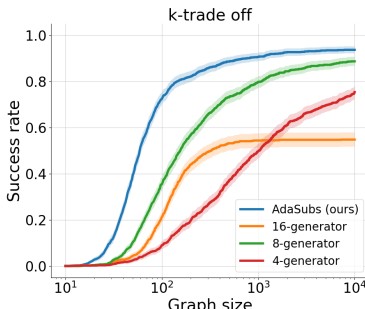

Figure 4: Comparison of success rates for different subgoal generators for Sokoban. AdaSubS-$k$ describes using a single generator with distance $k$.

when more compute is available. Crucially, AdaSubS with multiple generators successfully resolves the trade-off, outperforming every single generator, see Figure 4.

**Verifier: precision and recall** For each subgoal, the verifier outputs a classification probability $p$, which is used to accept the subgoal (if $p > t_{hi}$), reject the subgoal (if $p < t_{lo}$) or to pass to the further verification (if $p \in [t_{lo}, t_{hi}]$) by CLLP. For the acceptance task, we require high precision, as one false positive can lead to the failure of the whole search procedure. This leads to a high value for the corresponding parameter $t_{hi}$ (e.g. for Sokoban $t_{hi} = 0.99$, which corresponds to a precision rate of 97%). For the rejection task, we do not wish to reject true positives. In other words, we aim for high recall. In this case, errors usually increase the search cost; the corresponding parameter $t_{lo}$ is set to $t_{lo} = 0.1$ for Sokoban (this gives the recall of 99%). Additionally, for the verifier to be useful, we need to avoid passing subgoals to the further verification $p \in [t_{lo}, t_{hi}]$. It turns out that for the thresholds selected, the verifier is able to assess 70% states without the assistance of CLLP.

### G.6 Adaptive Subgoal Search flowchart

**Adaptive subgoal search**

## H  DEVELOPING ADAPTIVE SEARCH METHODS

There are many natural ways to incorporate adaptivity into the subgoal search pipeline. We experimented with several designs to find one that gives strong results in any domain. Here we provide a detailed description of all the variants tested and the numerical results of their evaluation in our environments. Their implementations can be found in Section H.2.

An adaptive algorithm should adjust the complexity of the proposed subgoals to the local complexity of the environment in the neighbourhood of the processed state. This can be realized using the following two approaches:

- Use an adaptive planner that provides a list of $k$-generators, the most promising node in every step selects and a generator to expand it.

- Use an adaptive subgoal generator that instead of proposing fixed-distance subgoals learns to automatically adjust the distance.

### H.1  ADAPTIVE PLANNERS

When implementing adaptivity with the planner, we need to specify a list of $k$-generators $\rho_{k_0}, \ldots, \rho_{k_m}$. In every iteration, the algorithm will select a node to expand and generators from the list that will create the new subgoals. This way, it can directly control the complexity of the subgoals and adapt to the current state and progress of the search.

**MixSubS.** Given a list of trained $k$-generators, a simple approach is to call all of them each time a node is expanded. In every iteration, we choose the node with the highest value in the tree and add subgoals proposed by each generator $\rho_{k_0}$ to $\rho_{k_m}$. See Algorithm 6 for the implementation.

Observe that in the easy areas of the environment the search will progress fast, since the furthest subgoal will most likely have the highest value, so it will be chosen as the next node to expand. On the other hand, in the hard parts the shortest generators are more likely to provide subgoals that advance towards the target at least a step.

This method already achieve superior results compared to single generators, both on small and large budget. In the Rubik environment, it even reaches $100\%$ solved cubes. MixSubS provides the advantage of planning with different horizons, but at the same time, it produces many unnecessary nodes in the easy areas, while taking only long steps is sufficient to solve the task. Additionally, one may want to prioritize the generators that perform better, which cannot be done with this method.

**Iterative mixing.** In this approach, we specify a number of iterations $l_i$ for each generator $\rho_{k_i}$. We use $\rho_{k_0}$ to expand the highest-valued nodes in the first $l_0$ iterations. Then, we use $\rho_{k_1}$ to expand the best nodes in the following $l_1$ iterations and the procedure follows for the consecutive generators. After finishing with the last one, we start again from the beginning. See Algorithm 7 for the implementation.

This algorithm offers the flexibility of specifying the exact number of iterations for each generator, which forms an explicit prioritization. It can resemble some of the listed algorithms for carefully chosen $l_i$. However, tuning the number of iterations requires much more effort than the other parameter-free algorithms do. Therefore, we experimented with another two mixing approaches that select the generator automatically in every iteration.

**Strongest-first.** Another natural implementation of the planner is to choose the node with the highest value and expand it with the longest generator that was not used there yet. See Algorithm 8 for the implementation. While this greedy approach maintain clear advantage over single generators, it is outperformed by most of the mixing methods, even the simple mixes. We hypothesize that this method is more sensitive to the errors of the value function – if the search enters an area that the value function estimates too optimistically, it spends too much time trying to exploit it.

**Longest-first (used by AdaSubS).** This method in every iteration selects the longest generator that has at least one node to expand and highest-valued node for that generator in the queue. This way, it explicitly prioritizes using the longest generators and turns to the shorter only when the search is stuck. See Algorithm 9 for the implementation. As shown in the tables below, this method outperforms all other designs, in all the environments and within all budget constraints. It prioritizes

the better generators, but does not require any additional hyperparameters to by specified. Therefore, we consider it the best mixing algorithm and use in AdaSubS as the default planner.

## H.2 ADAPTIVE PLANNERS IMPLEMENTATIONS

In this section we provide the implementations of the planners. The lines highlighted in blue indicate the differences with the AdaSubS code. All the methods require specifying the list of generators $\rho_{k_0}, \ldots, \rho_{k_m}$. The Iterative mixing planner additionally requires a list of iterations $l_0, \ldots, l_m$.

---

**Algorithm 6** MixSubS

**function** SOLVE($\mathbf{s}_0$)
    T $\leftarrow \emptyset$            ▷ priority queue
    parents $\leftarrow \{\}$
    T.PUSH($(V(\mathbf{s}_0), \mathbf{s}_0)$)
    seen.ADD($\mathbf{s}_0$)
    **while** $0 <$ LEN(T) and LEN(seen) $< C_1$ **do**
        \_, s $\leftarrow$ T.EXTRACT\_MAX()
        subgoals $\leftarrow \{\rho_{k_1}(\mathbf{s}), \ldots, \rho_{k_m}(\mathbf{s})\}$
        **for** s$'$ **in** subgoals **do**
            **if** s$'$ **in** seen **then** continue
            **if** not IS\_VALID($\mathbf{s}, \mathbf{s}'$) **then**
                continue
            seen.ADD($\mathbf{s}'$)
            parents[$\mathbf{s}'$] $\leftarrow s$
            T.PUSH($(V(\mathbf{s}'), \mathbf{s}')$)
            **if** SOLVED($\mathbf{s}'$) **then**
                **return** LL\_PATH($s'$, parents)
    **return** False

---

**Algorithm 7** Iterative mixing

**function** SOLVE($\mathbf{s}_0$)
    T$_{k_i} \leftarrow \emptyset$      ▷ $m+1$ priority queues
    parents $\leftarrow \{\}$
    **for** $k$ **in** $k_0, \ldots, k_m$ **do**
        T$_k$.PUSH($(V(\mathbf{s}_0), \mathbf{s}_0)$)
    seen.ADD($\mathbf{s}_0$)
    cnt $\leftarrow 0$        ▷ Iterations counter
    id $\leftarrow 0$        ▷ Current generator id
    **while** $0 <$ LEN(T) and LEN(seen) $< C_1$ **do**
        **if** cnt $= l_{\text{id}}$ **or** LEN($T_{k_{\text{id}}}$) $= 0$ **then**
            id $\leftarrow$ (id $+ 1$)%$(m+1)$, cnt $\leftarrow 0$
        cnt $\leftarrow$ cnt $+ 1$
        \_, s $\leftarrow$ T$_{k_{\text{id}}}$.EXTRACT\_MAX()
        subgoals $\leftarrow \rho_{k_{\text{id}}}(\mathbf{s})$
        **for** s$'$ **in** subgoals **do**
            **if** s$'$ **in** seen **then** continue
            **if** not IS\_VALID($\mathbf{s}, \mathbf{s}'$) **then**
                continue
            seen.ADD($\mathbf{s}'$)
            parents[$\mathbf{s}'$] $\leftarrow s$
            **for** $k$ **in** $k_0, \ldots, k_m$ **do**
                T$_k$.PUSH($(V(\mathbf{s}'), \mathbf{s}')$)
        **if** SOLVED($\mathbf{s}'$) **then**
            **return** LL\_PATH($s'$, parents)
    **return** False

---

**Algorithm 8** Strongest-first

**function** SOLVE($\mathbf{s}_0$)
    T $\leftarrow \emptyset$    ▷ priority queue with lexicographic order
    parents $\leftarrow \{\}$
    **for** $k$ **in** $k_0, \ldots, k_m$ **do**
        T.PUSH($((V(\mathbf{s}_0), k), \mathbf{s}_0)$)
    seen.ADD($\mathbf{s}_0$)
    **while** $0 <$ LEN(T) and LEN(seen) $< C_1$ **do**
        (\_, $k$), s $\leftarrow$ T.EXTRACT\_MAX()
        subgoals $\leftarrow \rho_k(\mathbf{s})$
        **for** s$'$ **in** subgoals **do**
            **if** s$'$ **in** seen **then** continue
            **if** not IS\_VALID($\mathbf{s}, \mathbf{s}'$) **then**
                continue
            seen.ADD($\mathbf{s}'$)
            parents[$\mathbf{s}'$] $\leftarrow s$
            **for** $k$ **in** $k_0, \ldots, k_m$ **do**
                T.PUSH($((V(\mathbf{s}'), k), \mathbf{s}')$)
        **if** SOLVED($\mathbf{s}'$) **then**
            **return** LL\_PATH($s'$, parents)
    **return** False

---

**Algorithm 9** Longest-first

**function** SOLVE($\mathbf{s}_0$)
    T $\leftarrow \emptyset$    ▷ priority queue with lexicographic order
    parents $\leftarrow \{\}$
    **for** $k$ **in** $k_0, \ldots, k_m$ **do**
        T.PUSH($((k, V(\mathbf{s}_0)), \mathbf{s}_0)$)
    seen.ADD($\mathbf{s}_0$)
    **while** $0 <$ LEN(T) and LEN(seen) $< C_1$ **do**
        ($k$, \_), s $\leftarrow$ T.EXTRACT\_MAX()
        subgoals $\leftarrow \rho_k(\mathbf{s})$
        **for** s$'$ **in** subgoals **do**
            **if** s$'$ **in** seen **then** continue
            **if** not IS\_VALID($\mathbf{s}, \mathbf{s}'$) **then**
                continue
            seen.ADD($\mathbf{s}'$)
            parents[$\mathbf{s}'$] $\leftarrow s$
            **for** $k$ **in** $k_0, \ldots, k_m$ **do**
                T.PUSH($((k, V(\mathbf{s}')), \mathbf{s}')$)
        **if** SOLVED($\mathbf{s}'$) **then**
            **return** LL\_PATH($s'$, parents)
    **return** False

## H.3 ADAPTIVE GENERATORS

A $k$-generator is trained to propose subgoals that should be exactly $k$ steps ahead. However, instead of matching a fixed distance, it can opt for long subgoals when the next steps are clear and short when difficulties appear, or both if it is not certain.

Implementing this idea requires changing the training of the generator. Given a training trajectory, for each state $s_i$ we need to select the target state $s_{t(i)}$ that should be the output of the generator. We tested a few methods that select this target.

**Longest-reachable** We use the low-level conditional policy to estimate the local complexity around $s_i$. Specifically, we choose $s_{t(i)}$ to be the furthest state on the trajectory such that it is reachable from $s_i$ with the CLLP and so do all its predecessors. In other words, we check whether CLLP starting in $s_i$ can reach $s_{i+1}$, $s_{i+2}$, etc. When we find the first state $s_j$ that is not reachable, we set $t(i)$ to be $j - 1$.

Intuitively, this approach makes the generator learn to output subgoals as distant as possible, but still reachable for CLLP. However, this way the targets are selected on the borderline of reachability, which may lead to too hard subgoals in some cases.

**Sampling-reachable** To make target state selection more robust, we modify reachability verification. Instead of greedily following the best action determined by CLLP probabilities, in every step we sample the action. This way, we are more likely to take suboptimal actions, so the selected target should be reachable with higher confidence.

**Secondary-reachable** Another method of making more robust selection is to follow the action with the lowest probability that exceeds a fixed threshold, e.g. $25\%$. Intuitively, we follow the action that CLLP considers to be good, but is less certain than in the case of the highest-ranked. Therefore, a subgoal reached in this way should be reachable with even higher confidence when following the greedy actions.

Our experiments show that the adaptive generators trained according to those designs perform well in the environments we consider. For instance, all the methods nearly reach a $90\%$ solve rate on Sokoban. However, none of them provide better results than the kSubS baseline. Therefore in this work we focus on planner-based adaptivity and leave tuning the adaptive generators pipeline for future work.

## H.4 BENCHMARKING RESULTS

Tables 10-12 show the numerical results achieved by the adaptive planners described in section H.1, compared to baselines: BestFS and kSubS. For some of the methods a few variants are provided. In each table, the longest-first, strongest-first and iterative mixing methods use the same set of generators: $[3, 2, 1]$ for INT, $[4, 3, 2]$ for Rubik, and $[8, 4, 2]$ for Sokoban. Our main algorithm, Adaptive Subgoal Search, uses the longest-first planner and the verifier network.

| | | INT | | | |
|---|---|---|---|---|---|
| | | Small budget (50 nodes) | | Large budget (1000 nodes) | |
| | | with verifier | without | with verifier | without |
| BestFS | | - | 1.7% | - | 36.7% |
| kSubS | k=4 | 2.2% | 0.1% | 82.4% | 83.0% |
| | k=3 | 4.0% | 0.2% | 89.6% | 90.7% |
| | k=2 | 2.1% | 0.5% | 89.8% | 91.7% |
| | k=1 | 0.0% | 0.0% | 34.7% | 46.0% |
| MixSubS | k=[4,3,2] | 0.0% | 0.0% | 94.6% | 95.0% |
| | k=[3,2,1] | 0.0% | 0.0% | 92.2% | 92.9% |
| | k=[3,2] | 17.0% | 14.8% | 92.2% | 93.5% |
| Iterative mixing | iterations=[1,1,1] | 32.0% | 30.1% | 87.0% | 88.6% |
| | iterations=[10,1,1] | 43.0% | 44.8% | 95.1% | 96.0% |
| | iterations=[4,2,1] | 54.0% | 52.1% | 93.6% | 95.5% |
| Strongest-first | | 39.5% | 40.8% | 88.5% | 89.8% |
| Longest-first (AdaSubS) | | 59.0% | 51.5% | 95.7% | 95.5% |

Table 10: INT benchmark

| | | Rubik | | | |
|---|---|---|---|---|---|
| | | Small budget (400 nodes) | | Large budget (6000 nodes) | |
| | | with verifier | without | with verifier | without |
| BestFS | | - | 0.0% | - | 1.8% |
| kSubS | k=4 | 28.8% | 24.5% | 98.6% | 98.8% |
| | k=3 | 19.3% | 18.6% | 95.6% | 95.4% |
| | k=2 | 8.2% | 4.5% | 99.0% | 95.8% |
| | k=1 | 0.5% | 0.5% | 76.5% | 76.5% |
| MixSubS | k=[4,3,2] | 29.1% | 20.9% | 99.1% | 100.0% |
| | k=[4,3] | 49.1% | 45.1% | 99.2% | 100.0% |
| Iterative mixing | iterations=[1,1,1] | 33.5% | 23.0% | 99.2% | 100.0% |
| | iterations=[10,1,1] | 50.6% | 43.6% | 99.1% | 99.9% |
| | iterations=[4,2,1] | 48.4% | 41.2% | 99.2% | 100.0% |
| Strongest-first | | 33.4% | 27.1% | 99.0% | 99.9% |
| Longest-first (AdaSubS) | | 58.0% | 52.4% | 99.2% | 100.0% |

Table 11: Rubik benchmark

| | | Sokoban | | | |
|---|---|---|---|---|---|
| | | Small budget (100 nodes) | | Large budget (5000 nodes) | |
| | | with verifier | without | with verifier | without |
| BestFS | | - | 45.9% | - | 82.6% |
| kSubS | k=16 | 13.7% | 5.1% | 60.5% | 63.5% |
| | k=8 | 26.0% | 4.7% | 85.6% | 84.4% |
| | k=4 | 8.2% | 2.6% | 68.1% | 65.5% |
| | k=2 | 1.4% | 0.7% | 40.0% | 38.3% |
| MixSubS | k=[8,4,2] | 52.7% | 37.7% | 91.7% | 90.2% |
| | k=[16,8,4] | 55.6% | 44.9% | 89.1% | 89.0% |
| Iterative mixing | iterations=[1,1,1] | 52.7% | 37.7% | 91.7% | 90.2% |
| | iterations=[10,1,1] | 68.3% | 58.6% | 92.5% | 92.1% |
| | iterations=[4,2,1] | 64.5% | 52.6% | 93.5% | 93.2% |
| Strongest-first | | 54.6% | 41.9% | 92.0% | 90.8% |
| Longest-first (AdaSubS) | | 72.2% | 63.4% | 93.4% | 93.6% |

Table 12: Sokoban benchmark

## I   Optimality of solutions for Sokoban

We did an additional experiment for Sokoban: we used simple BFS to find the optimal path, and we compared its length with the outcome of AdaSubS. The results are as follows (tested on 600 Sokoban boards):

- For 7% of boards, AdaSubS found the optimal solution.
- The average difference between the length of the AdaSubS solution and the optimal solution is 15 steps.
- On average, the AdaSubS solution is 38% longer than the optimal one.

## J    INFRASTRUCTURE USED

We performed experiments using two types of hardware: with and without access to GPUs. In the former, we used nodes equipped with a single Nvidia V100 32GB card or Nvidia RTX 2080Ti 11GB card. Each such node had 4 CPU cores and 168GB of RAM. In the latter, we used nodes equipped with Intel Xeon E5-2697 2.60GHz CPU with 28 cores and 128GB RAM.

Each transformer model was trained on a single GPU node for 3 days. Sokoban models were trained on CPU nodes (due to the small size of the models).

## K  OUT-OF-DISTRIBUTION ANALYSIS

In Rubik's Cube, we trained the generators on very short trajectories, obtained by applying only 10 random moves to the ordered cube (that covers about $7 \cdot 10^9$ configurations, compared to $4 \cdot 10^{19}$ in total). Even with such limited data, AdaSubS succeeded in solving 99% of the fully scrambled cubes we tested.

In Sokoban, we check that AdaSubS trained on boards with 4 boxes can tackle instances with 5, 6, and 7 boxes. Specifically, it solves 87%, 82%, and 74% of cases, respectively, while kSubS only solves 74%, 67%, and 56%, respectively. The results are presented in Figure 5. Note that exactly like in the case of INT, AdaSubS consistently has half the failure rate of that of kSubS on OOD instances.

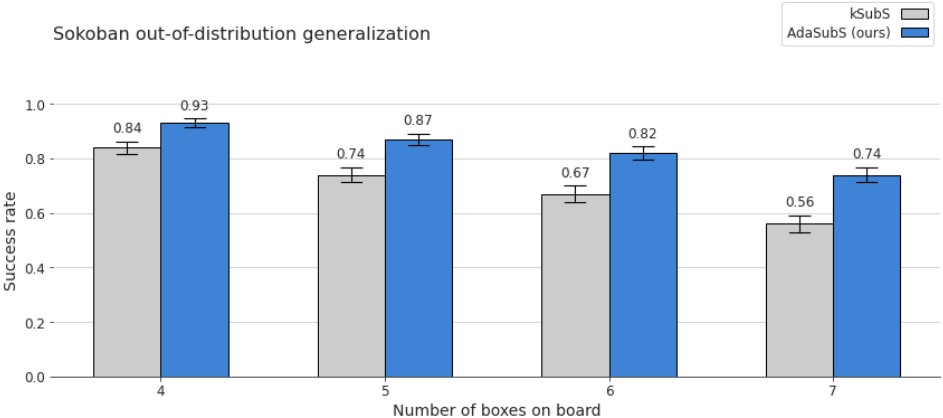

Figure 5: Out-of-distribution performance of AdaSubS and kSubS for Sokoban boards with more boxes, with budget of 5000 nodes. Both methods were trained on instances with 4 boxes. Error bars correspond to 95% confidence intervals.

