# OpenReview forum: "Fast and Precise: Adjusting Planning Horizon with Adaptive Subgoal Search"
_ICLR.cc/2023/Conference — ICLR 2023 notable top 5%_

### Official Review · Reviewer_1BT4 · 2022-10-21

**Confidence:** 4
**Correctness:** 4
**Technical Novelty And Significance:** 3
**Empirical Novelty And Significance:** 3
**Recommendation:** 8

**Clarity, Quality, Novelty And Reproducibility:**

- [Clarity & Quality] the paper is very well written. There are a few gramatical issues (some mentioned below), but otherwise the paper is very easy to follow. The overall approach is well-motivated and clear. Moreover, the appendices are rich with detail and will be no doubt helpful for those who which to implement the approach for themselves.
- [Novelty] The paper grows the state of the art . The approach is novel, though the amount of novelty is not particularly significant.
- [Reproducibility] The authors have included all source code and documentation. The paper appears to be fully reproducible.

**Strength And Weaknesses:**

High-level comments
- [strong] The paper improves upon the state of the art. Search finds the goal in problem challenging problem instances including longer INT problems than are feasible for other approaches in this space. The results are solid and the analysis is comprehensive. This is a central strength of the paper.
- [strong] The choice of experiments is good and appropriate for the work.
- [weak] There is no discussion of optimal paths and the quality of the generated paths between the different approaches. For the INT domain, this is perhaps not possible, but for the other two domains, it would be very helpful to understand if the proposed approach improves success at the expense of plan quality, and the extent to which this tradeoff is made. While I do not believe inclusion of this discussion is essential, the paper would be improved by the inclusion of such a discussion.
- [weak] Sec. 4.5 (analyzing the quality of individual components) seems to omit a key piece of analysis: how well AdaSubS will perform without the verifier or how well the kSubS baseline will perform with the addition of the verifier. The central distinctions between the two strategies are (1) one versus multiple subgoal generators and (2) the inclusion of the verifier network. I do not believe there are any experiments that seek to understand how impactful the verifier. It could therefore be the case that the low level policy network is not particularly capable and that all of the performance improvement comes from the addition of the verifier, which bypasses that process. Some discussion or additional experiments (or a pointer to where these experiments exist) would be incredibly helpful for understanding.

Smaller comments, questions, and typos:
- Abstract: Though the abstract makes sense after having read the paper, the first couple sentences are not particularly clear on their own. Consider revising those to provide a clearer picture of what the central, unique insight is.
- Sec. 1: The figure in the introduction, while informative, is perhaps somewhat misleading, since robot motion is not studied in this work. Perhaps it would be more appropriate to use the Sokoban domain instead.
- Sec. 2: Typo (I believe) "variable environment variable complexity".
- In Sec. 3, the discussion of the thresholds could be clearer; it was not clear to me until later on that there were two thresholds.
- Algorithm 1: The 'return' statement may not find a path, in which (I believe) the procedure is to keep looping through paths if a feasible path is not found.
- Algorithm 1: Also, the first 'for' loop that pushes elements into T appears erroneous. There are not yet multiple values of 'k', since the subgoal network has not been queried; should this simply use k=0 and push the starting state into T?


**Summary Of The Paper:**

This paper presents Adaptive Subgoal Search (AdaSubS), an approach to learning-driven search that uses proposed subgoals and a novel "verifier" network to make more direct and rapid progress towards the target state. The approach builds upon recent progress in this space, notably the BF-kSubS, by proposing subgoals thought to be reachable a distance "k" from the current state towards the goal (for multiple values of k). Combining this multi-distance subgoal generation with a verifier network to prune subgoal not expected to be easily reachable, the approach makes faster progress towards the goal and is more successful at solving state-space search problems than competitive baselines. The authors study a Sokoban and Rubik's Cube domain and "INT" a theorem proving domain.

**Summary Of The Review:**

This is a solid paper with a somewhat small theoretical contribution bolstered by impressive results in a challenging domain and clear and comprehensive analysis. There are still a few open questions that I believe the paper should address pertaining to the importance of a few of the components and comparison to baselines.

---

> ### Author Response · Authors · 2022-11-19
> **Answer to the Reviewer 1BT4**
>
> We thank the Reviewer for the useful suggestions and we are happy to see that the Reviewer acknowledges the solidity of our results. Regarding the questions raised:
>
> - There is no discussion of optimal paths and the quality of the generated paths between the different approaches.
>
> The scope of this paper is to develop a domain-agnostic and efficient search method (in terms of the used computation budget). We do not seek optimality, and this lets us relax various assumptions, including using suboptimal (=easy to get) data. We mention this in the Limitation section.
>
> However, we agree that the optimality of the paths is an interesting and important metric to consider. We did an additional experiment for Sokoban: for 600 problem instances, we used the graph searching algorithm BFS to find the optimal path and we compared its length with the outcome of AdaSubS. Despite we don’t focus on the optimality of the solutions, AdaSubS finds paths that are reasonably close to optimal. Specifically:
>
> 1. For 7% of boards, AdaSubS found the optimal solution.
> 2. The average difference between the length of the AdaSubS solution and the optimal solution is 15 steps.
> 3. On average, the AdaSubS solution is 38% longer than the optimal one.
>
> These results are described in Appendix K and referred to in Section 4.3.
>
> - Sec. 4.5 (analyzing the quality of individual components) seems to omit a key piece of analysis: how well AdaSubS will perform without the verifier or how well the kSubS baseline will perform with the addition of the verifier
>
> In Table 1 (the main text) and Tables 10, 11, and 12 (Appendix) we present a detailed comparison of different algorithms, including the performance of kSubS (for different k) with and without a verifier, and AdaSubS with and without a verifier (AdaSubS is described in the tables as longest-first). To summarize the results: the advantage of using a verifier is particularly visible on small budgets for both kSubS and AdaSubS. We observe the most significant improvement in Sokoban, where we use the longest subgoals. For large budgets this gain is diminished, but is still justified as the properly tuned verifier never decreases the results significantly. We put a short note regarding this in Sec 4.4.
>
> - It could therefore be the case that the low level policy network is not particularly capable and that all of the performance improvement comes from the addition of the verifier, which bypasses that process.
>
> In practice, our CLLP has a 95%- 100% success rate in reaching correct subgoals. The verifier is trained to predict whether the policy can reach a given subgoal, so its only role is to decrease the computation cost of running CLLP. Moreover, if the verifier accepts a subgoal that CLLP can't reach, the solver may fail to provide the solving path (see Appendix A).
>
> - Sec. 1: The figure in the introduction, while informative, is perhaps somewhat misleading, since robot motion is not studied in this work. Perhaps it would be more appropriate to use the Sokoban domain instead.
>
> We are aware that the current version of the figure might feel somewhat misleading. We held an internal discussion and found that the current version is the best in terms of intuitive readability: most people can relate to driving a car, covering substantial distances “on autopilot” (which corresponds to farther subgoals), and being focused when the situation on the road demands attention (closer subgoals). A few alternative versions of the figure using Sokoban turned out to be hard to grasp and required a detailed description, including the game's rules. We thus suggest keeping the image or omitting it at all.
>
> - Algorithm 1: The 'return' statement may not find a path, in which (I believe) the procedure is to keep looping through paths if a feasible path is not found.
>
> In a setup without the verifier, each subgoal that is added to the tree is guaranteed to be reachable by CLLP, and LL_PATH always succeeds. If the verifier is used, it might happen that it could wrongly classify a subgoal as correct. If such a subgoal appears on the final path, we have to declare the problem as unsolved. Note that this is a rather rare case, as we intentionally tune $t_{hi}$ to relatively high values, and the errors are benign if they happen outside of the solution path. For example, in Rubik’s cube, only 0.8% of cases fail due to such errors.
>
> We also note that in practice, we use caching to lower the number of calls to CLLP, which is now explained in Algorithm 4.
>
> - In Sec. 3, the discussion of the thresholds could be clearer; it was not clear to me until later on that there were two thresholds.
>
> We agree, the clarification has been added.

---

> > ### Author Response · Authors · 2022-11-19
> > **Answer to the Reviewer 1BT4 continued**
> >
> > - Algorithm 1: Also, the first 'for' loop that pushes elements into T appears erroneous. There are not yet multiple values of 'k', since the subgoal network has not been queried; should this simply use k=0 and push the starting state into T?
> >
> > T is a queue of nodes that can be potentially used as queries to the generators. If a tuple (k, V(s), s) is retrieved from the queue, the algorithm generates subgoals from s with the k-generator (see the second line of the while loop in Algorithm 1). In the first ‘for’, we initialize the queue so that we start from the initial state and allow to use of any generator.

---

> > > ### Author Response · Authors · 2022-11-24
> > > **Gentle ping**
> > >
> > > Please let us know if you are satisfied with our answers and if there are any issues left.

---

### Official Review · Reviewer_1Tzk · 2022-10-25

**Confidence:** 4
**Correctness:** 4
**Technical Novelty And Significance:** 3
**Empirical Novelty And Significance:** 3
**Recommendation:** 8

**Clarity, Quality, Novelty And Reproducibility:**

The writing is sometimes too concise, leaving many implementation and experiment details in the appendix. The method and experiment design are novel. It's great that the authors have provided links to their codebase, which enables researchers to validate their approach and build upon their work.

**Strength And Weaknesses:**

The idea of learned subgoal generators with adaptive planning horizons is novel. The authors performed large-scale experiments on hard planning problems, comparing their methods both with baselines and also on out-of-distribution problems. That's very good efforts.

Before authors' rebuttal, the paper lacks important details in the main text for understanding. Though the authors stated the size of their training dataset. It's unclear how many problems were used for the results from Figure 2 and Figure 3. It would be helpful to state that in section 4.1 to make the results more convincing. It's also unclear whether the total wall time in Table 2-4 in the Appendix is from solving one problem or multiple problems, and how they compare to BestFS baseline. Without context, 3-26 hours seems too long for the title of the paper to be called "fast and precise."

**Summary Of The Paper:**

The authors proposed an algorithm for planning that uses learned components for generating subgoals (states up to k steps away), generating one low-level action between two states (conditional low-level policy, CLLP), verifying whether CLLP can find an action between two states, and predicting the distance between current state and goal state. The novelty from previous work (kSubS) is that multiple subgoal generators are used to adapt the planning horizon based on states. The modules are trained on offline datasets of puzzles with a large state space. Experiment results show that their algorithm is able to solve the problem of visiting fewer states.

**Summary Of The Review:**

Overall it's an exciting paper, tackling hard planning problems with a novel learning-based algorithm. With more details provided about their evaluation, it would make a very strong submission.

---

> ### Author Response · Authors · 2022-11-19
> **Answer to the Reviewer 1Tzk**
>
> We thank the Reviewer for an encouraging review and acknowledging novelty.
>
> - It's unclear how many problems were used for the results from Figure 2 and Figure 3. It would be helpful to state that in section 4.1 to make the results more convincing.
>
> In each experiment, we used 1000 instances of a given problem to evaluate our algorithm. We now point it out explicitly in Sec 4.2, which is perhaps a slightly better place than the suggested Sec 4.1.
>
> - It's also unclear whether the total wall time in Table 2-4 in the Appendix is from solving one problem or multiple problems
>
> Following the suggestions, we’ve changed the tables in the appendix to report the average number of calls and the average wall time (the average time span is between 10 and 100 seconds). We hope that this will make the interpretation of our results easier.
>
> We have included the Reviewers' comments and made multiple other improvements (marked in orange in the pdf). These, we think, significantly improve the quality of the paper. We, thus, gently ask the Reviewer to consider increasing the score. At the same time, we would be more than happy to address any further suggestions.

---

> > ### Comment · Reviewer_1Tzk · 2022-11-21
> > **I acknowledge the improvements by increasing the score**
> >
> > Thank you for responding to my comments and adding those details. With the other revisions, the paper's contributions is now clearer to me. I agree with strongly accepting the paper.

---

### Official Review · Reviewer_EUhR · 2022-10-26

**Confidence:** 4
**Correctness:** 3
**Technical Novelty And Significance:** 3
**Empirical Novelty And Significance:** 2
**Recommendation:** 8

**Clarity, Quality, Novelty And Reproducibility:**

The work is well-written and clear. The experimental settings are well explored and
the comparative analysis of the different algorithms gives a better understanding of
the performance on the complex domains. The two main blocks of the algorithm (the
sub-goal generator and the verifier network) are inspired by the existing works which
are clearly mentioned in the related work section.


**Strength And Weaknesses:**

Strengths:
● The paper is well-structured and easy to read.
● Introduction of novel search algorithm benefiting from both the planning and
learning methods.
● The experimental section is very well organized and provides good insights into
the complexity of the domains considered.
● State-of-the-art results on inequality theorem prover INT.

Weakness:
- Although the paper shows advancement in learning-based planning, the results are not surprising given that there is a large body of work in search-based planning to show that adaptive action selection / heuristic selection is beneficial in planning. See work like "Real-Time Adaptive A*" or historical perspective in a planning book like Ghallab et al. See also recent dissertation titled "Adaptive Search Techniques in AI Planning and Heuristic Search" by Maximilian Fickert.
- A problem with this like of work is that the algos are not complete. They are not guaranteed to  always produce a plan even if such a plan exists. This point should be highlighted.
* A system architecture diagram depicting the flow of the algorithm would have
been more comprehensible. It is hard to understand if all the chosen sub-goals
at a given node are being explored.
● It is mentioned that the verifier network is a binary classification model, and
if the network fails to predict the feasibility of the sub-goal, the algorithm
falls back on to CLLP to decide whether to keep or discard a given subgoal. But
it is not clear how CLLP is used for this purpose as it generates an action
plan between two given states. Does an empty sequence from CLLP mean the
sub-goal is being rejected?
● The authors commented on the out-of-distribution generalizability of the
inequality theorem prover INT, however, it could have been interesting to see
the status of the other domains.

**Summary Of The Paper:**

The paper introduces a new algorithm Adaptive Subgoal Search (AdaSubS), which aims to generate the prospective sub-goals iteratively to generate a good action plan. The system has innovative components of a subgoal generator, Conditional low-level policy (CLLP) to find action steps and a verification algorithm to swiftly prune the faulty sub-goals obtained.

The authors present comprehensive results to show that adaptive methods work.


**Summary Of The Review:**

The paper introduces a new algorithm called Adaptive Subgoal Search (AdaSubS). It covers related work and provides extensive experimental results provided which give an overall insight into the performance of the algorithm on the different problems chosen. However, the paper needs to add details justifying the need for multiple blocks for sub-goal verification and contextualize the properties of the planner. The entire flow of the algorithm needs to be summarized in a flow chart format. The memory and time usage details of the algorithm needs to be mentioned.

The paper should also connect with well known results in search-based planning on adaptive action selection and heuristic selection. Otherwise, it is re-discovering known behaviors in a new setting.

---

> ### Author Response · Authors · 2022-11-19
> **Answer to the Reviewer EUhR**
>
> We thank the Reviewer for appreciating our algorithm and experiments and for the valuable remarks on the content.
>
> - Although the paper shows advancement in learning-based planning, the results are not surprising given that there is a large body of work in search-based planning to show that adaptive action selection / heuristic selection is beneficial in planning.
>
> The idea of using adaptive techniques is well-known indeed. However, there is a gap between expectations and how to achieve them. Our work bridges this gap: we showed how to achieve adaptivity and why it works. We comprehensively evaluated a spectrum of natural implementations and selected the one that consistently performs best across all tested environments (AdaSubS). Interestingly, there were some surprises along the way; most notably and counterintuitively, initially the most promising candidate, MixSubS, performed sub-optimally.
>
> Thank you for providing us with the relevant literature on adaptivity, we included Fickert (2022) in the related works.
>
> - A problem with this line of work is that the algos are not complete. They are not guaranteed to always produce a plan even if such a plan exists.
>
> With a slight modification, AdaSubS can be guaranteed to find a solution to any given problem, provided a large enough computational budget. We achieve that by adding an exhaustive single-step policy as the last generator, which would populate an empty queue with all children of the highest valued not yet expanded node. In practice we didn’t find it necessary to obtain strong results. We added that remark in Section 3 and Appendix G.1. However, we admit that this is a limitation of our hierarchical search, so we added that point to the limitations section.
>
> - A system architecture diagram depicting the flow of the algorithm would have been more comprehensible.
>
> We agree with the suggestion. Therefore, we updated the diagram in Figure 1 to improve its clarity.
>
> - CLLP decides whether to keep or discard a given subgoal, but it is not clear how CLLP is used for this purpose.
>
> When we generate a subgoal s' for state s, we first validate it with the verifier network (see Alg 3). In case it is uncertain, we generate the action plan between s and s' with CLLP (in the last line of Alg 3 GET_PATH is called). Iteratively, we query the CLLP to predict the next action, execute it to observe a new state, and repeat until we reach s' or exceed the limit of steps (see Algorithm 2). If we succeed, the subgoal s' is declared valid. Otherwise, we return an empty list of actions to indicate its invalidity. We added the relevant comments in Algorithm 2.
>
> - The authors commented on the out-of-distribution generalizability of the inequality theorem prover INT, however, it could have been interesting to see the status of the other domains.
>
> We ran additional experiments to check how AdaSubS generalizes to out-of-distribution data in the other environments.
>
> In Rubik's Cube, we trained the generators on very short trajectories, obtained by applying only 10 random moves to the ordered cube (that covers about $7 \cdot 10^9$ configurations, compared to $4 \cdot 10^{19}$ in total). Even with such limited data, AdaSubS succeeded in solving 99% of the fully scrambled cubes we tested.
>
> In Sokoban, we check that AdaSubS trained on boards with 4 boxes can tackle instances with 5, 6, and 7 boxes. Specifically, it solves 87%, 82%, and 74% cases, respectively, while kSubS only 74%, 67%, and 56%, respectively. Observe that just like in the case of INT, AdaSubS consistently has twice a lower failure rate than kSubS on OOD instances. We have included these results in Appendix K.
>
> - Justification for multiple blocks
>
> The separate networks with separately trained parameters may act as an ensemble. Such a setup decouples the times when components make errors and introduces a compensation mechanism. For instance, if a value function is good, it should guide the search in the right direction, despite weak subgoal generators. Conversely, if the generators work well, most subgoals should offer progress, even for a weak value function.
>
> - The memory and time usage details of the algorithm needs to be mentioned.
>
> We provide an extensive analysis of running time in Appendix C (referred in Section 4). To improve its clarity, we now report the average statistics per episode. Additionally, we extended it with a discussion of memory usage details.
>
> Our algorithm keeps track of the search tree composed of high-level nodes. Thus, the amount of required memory grows linearly with the search budget. The BestFS baseline, which uses only low-level steps, usually requires much larger memory because it must record every step. In practice, when evaluating the BestFS baseline in the INT environment, we often had problems with experiments crashing because of exceeding the memory limit on the machine. We never observed similar issues when running AdaSubS.

---

> > ### Author Response · Authors · 2022-11-19
> > **Answer to the Reviewer EUhR continued**
> >
> > We hope our explanations are helpful and the changes in the paper improve its quality. We, thus, gently ask the Reviewer to raise the score or point out further shortcomings, if present.

---

### Official Review · Reviewer_Kkdv · 2022-11-04

**Confidence:** 5
**Correctness:** 3
**Technical Novelty And Significance:** 3
**Empirical Novelty And Significance:** 4
**Recommendation:** 8

**Clarity, Quality, Novelty And Reproducibility:**

The work is novel and insightful. The significance can be lower if the algorithm were harder to configure. The interaction of the attempted lookahead with the subgoal generation is interesting. The paper is clear enough, except for what I commented above.

Now I discuss at length some connections with related literature.
(This aspect is a weak point of the manuscript)

# Scholarship on search and planning

This is, I think, the 2nd time I see this paper submitted. I'm glad about the changes. I finally have an interpretation of what's going on. The unpacking of the algorithms in the appendix really helps. In general, I'm not very interested in the verifier. The interesting part is the depth.

Let me first discuss multi-queue in comparison with search algorithms used for planning.

First, PDDL-based planners are not comparable with the algorithms proposed here because of two reasons:
- They don't have the chance to train. Instead, they recieve the problem in PDDL, a high-level language for planning. For the planner, it's the first time they see that problem. Rubik's cube and Sokoban could be expressed there.
- Hyper-parameter tuning is only about the search and heuristics. Moreover, the evaluation tends to focus on finding the parameters that would solve multiple domains. PDDL-based planners intend to be autonomous tools, not human tuning involved.
- So, PDDL-based planners offer a lower bound on performance.
	- I don't think it's necessary to review those numbers here, but I'd expect so in a journal version of the paper.

Second, work on heuristic search –like Korf cited in the paper– it's the opposite of domain-independent. The heuristics are heavily adapted to the domain. So, they are not comparable but they offer an upper bound for learning-based algorithms.

Now, IterativeMixing is similar to alternation queues used in heuristics search algorithms used for PDDL-based planners.

That was first proposed by LAMA.
- Richter, Silvia, and Matthias Westphal. “The LAMA Planner: Guiding Cost-Based Anytime Planning with Landmarks.” Journal of Artificial Intelligence Research 39 (September 21, 2010): 127–77. https://doi.org/10.1613/jair.2972.

Nowadays, LAMA is usually tested as a configuration of the very flexible Fast Downward.
- Helmert, M. “The Fast Downward Planning System.” Journal of Artificial Intelligence Research 26 (July 12, 2006): 191–246. https://doi.org/10.1613/jair.1705. Sect 6.4 Multi-Heuristic Best-First Search
	- > "As an alternative to greedy best-ﬁrst search, Fast Downward supports an extended algorithm called multi-heuristic best-ﬁrst search. This algorithm differs from greedy best-ﬁrst search in its use of **multiple heuristic** estimators, based on our observation that different heuristic estimators have different weaknesses. It may be the case that a given heuristic is sufﬁcient for directing the search towards the goal except for one part of the plan, where it gets stuck on a plateau. Another heuristic might have similar characteristics, but get stuck in another part of the search space.
	- > "Various ways of combining heuristics have been proposed in the literature, typically adding together or taking the maximum of the individual heuristic estimates. We believe that it is often beneﬁcial not to combine the different heuristic estimates into a single numerical value. **Instead, we propose maintaining a separate open list for each heuristic estimator, which is sorted according to the respective heuristic. The search algorithm alternates between expanding a state from each open list**. Whenever a state is expanded, estimates are calculated according to each heuristic, and the successors are put into each open list.

Fast downward is very flexible:
- https://www.fast-downward.org/PlannerUsage
- https://www.fast-downward.org/Doc/SearchEngine

Other search engines for planning also incorporate multiple queues.
- https://lapkt-dev.github.io/docs/modules/

Actually, LAPKT has priorities over queues:
- https://github.com/LAPKT-dev/LAPKT-public/blob/d54b68fcc67d5d75ea40b3921220c29f49c71814/include/aptk/at_bfs_dq_mh.hxx#L353

Some planner "preferred operators", choose first a queue that tends to have fewer notes. If that's empty, they use the other one.

**That's similar to what AdaSubS is doing, as it always tries to expand over more steps first.**

Others algorithms can also be seen from this perspective.
Strongest-first  behaves more like selecting from the queue, breaking by k. (Can actual V values be the same for two different nodes?)
Longest-reachable becahse more hierarchical, like options.

## Some references on Sokoban and Rubik's cube

- Sokoban: https://github.com/AI-Planning/pddl-generators/tree/master/sokoban
- npuzzle, simpler to understand: https://github.com/AI-Planning/pddl-generators/tree/master/npuzzle
	- Rubik is a variation of that one. It's a challenging problem for planning
- A Comparison of Abstraction Heuristics for Rubik's Cube. Clemens Büchner, Patrick Ferber, Jendrik Seipp, Malte Helmert. https://icaps22.icaps-conference.org/workshops/HSDIP/
	- Uses pattern DBs, like use by Korf, but as planning
	- > None of the heuristics is able to solve problems where the optimal solution requires more than 13 rotations. Ac- cording to Rokicki et al. (2014), this restriction allows us to solve only 0.0001% out of the 4.3 · 1019 possible states of Rubik’s Cube. In comparison, Korf (1997) was able to optimally solve tasks at least 18 rotations away from the goal.1 The set of initial states with optimal cost 18 or less makes up approximately 98% of all possible initial states for Rubik’s Cube (Rokicki et al. 2014).
- Büchner, Clemens. “Abstraction Heuristics for Rubik’s Cube,” n.d., 44.
	- See here is a demo on the 3x3x3 (seems to be an undergrad project)
		- Muppasani, B., Pallagani, V., Lakkaraju, K., Srivastava, B., & Agostinelli, F. (2022). Solving the Rubik’s Cube with a PDDL Planner.
		- https://www.researchgate.net/profile/Bharath-Muppasani/publication/362555735_Solving_the_Rubik's_Cube_with_a_PDDL_Planner/links/62f13f8388b83e7320bb63c0/Solving-the-Rubiks-Cube-with-a-PDDL-Planner.pdf
	- Rubik's 2x2x2: https://wu-kan.cn/2019/11/21/Planning-and-Uncertainty/

I suggest the authors participate in the Planning and Learning competition that was just announced. https://ipc2023-learning.github.io

**Strength And Weaknesses:**

# Strengths

I'll be short here. See discussion below on my interpretation of the results.

- The paper is mostly well-organized and well written
- The ablation study confirms the role of the specific modules.
	- I appreciated the detailed algorithms in the appendix
- The conclusions are very clear, at least for the domains and conditions where it was tested.

# Weaknesses

- Scholarship: I think that Czechowski et al. (2021) should be mentioned as a starting point. The manuscript seems timid about it. I think a stronger paper would make that clear. Here is two key quotes:
	- > "kSubS is the first general learned hierarchical planning algorithm shown to work on complex reasoning domains Czechowski et al. (2021) (called BF-kSubS there), attaining strong results on Sokoban and Rubik’s Cube, and INT. kSubS can be **viewED** as a non-adaptive version of AdaSubS realized by a suitable hyperparameters choice: a single subgoal generator and inactive verifier (t lo = 0,t hi = 1)."
	- > "Most of the hyperparameters, both for training and evaluation, follow from Czechowski et al. (2021). The most important parameter of AdaSubS is the set of k-generators to use and the number of subgoals each of them generate.
	- Czechowski et al. (2021) introduced the idea of a search to achieve the goal in k steps. The paper has multiple references on how they used the same parameter.
- Scholarship: the paper does not review the state of the art on this problem. If the state of the art is not in the same scope, then related work should be discussed anyway.
	- For instance, see Cameron et al, IJCAI 2021, below.
- BestFS is not explained well enough
	- (See before my comment on contrib 2 below)
	- Given this is the most important baseline, the paper must make sure that the algorithms was properly tuned.
	- It seems it's a simple modification respect to the others, but perhaps there are hyperparameters that would work better for BestFS in comparison with AdaSubS
- No limitations are discussed (see my first question below as an instance)

# Questions: PLEASE ANSWER THESE ONES

- Does the algorithm keep a list of close nodes?
	- Do you cache the result of evaluations in Alg 1 or Alg 2?
	- It seems Alg 2 migh run predictions multiple times on the same nodes.
	- If a node is visited with k = 8, would it also be visited by GET_PATH with k = 2?
- How would AdaSubS behave in the following case?
	- Suppose a degenerated search space where
		- There are two action applicable actions in each state, but at least one of them always leads to a dead end.
		- Suppose further that there is actually only one path to the goal.
		- For a state in the path to the goal, there are two childs: one to the next state in the plan, the other one child that leads to a dead end by a short binary subtree, all of the deadends.
	- Suppose the same list of k used in Sokoban: [8, 4, 2]
	- Suppose that all the models are not very good. They choose uniformly over their options. Let's disable the verifier.
	- It seems that AdaSubS would expand multiple times the nodes in the subtrees that lead to deadends? Moreover, Alg 2, GET_PATH doesn't seem to have any memory. So, if the models are expensive this could grow quickly.
	- (If this example doesn't work, please attempt to provide another example where AdaSubS might deteriorate)
- Would you comment on the following paper published in IJCAI 2021?
	- Efficient Black-Box Planning Using Macro-Actions with Focused Effects. Cameron Allen, Michael Katz, Tim Klinger, George Konidaris, Matthew Riemer, Gerald Tesauro. https://arxiv.org/abs/2004.13242
	- Macro-actions are sequences of low-level actions. They create them by attempting to change a few variables in the problem.
	- They test in Rubik's cube.
	- They tested reusing macros in the same domain.
- Why does the evaluation support the 2nd main contribution?
	- End of sect 1 says: "We present a comprehensive study of adaptive methods, showing that they typically outperform non-adaptive ones". But given the lack of comments about other potential algorithms, this might not be true in general.
	- I'd accept something like "they outperform similar algorithms without the adaptation". It seems that the argument is that given that BestFS and the studied algorithms share common elements, then the value of the contributions is justified.
- How were the hyperparameters set?
	- I see the values in appendix F, but I don't see how they were set. Page 19:
		- > Based on experimental results, we have chosen generators of 8, 4, and 2 steps for Sokoban, 4, 3, and 2 steps for the Rubik’s Cube, and 4, 3, 2, and 1 step for INT. In the first two domains, the longest generator match the one used for kSubS.
	- Explaining that would help readers to understand the effort of using the algorithms in another setting.
	- Please clarify the computation cost of arriving at these hyperparameter values? For instance, how expensive is to realize that a set of hyperparameter values are not the most adequate?
- Are the hyper-parameters of BestFS set in the best possible way?
- Would you mention the multi-queue algorithms in a revised version of the paper? Where? (See that in my section "Scholarship on search and planning")

# Other changes

- Please mention the number of subgoals generators and the values of k early in the paper.
	- Table 1 shows that the list k is one of the most important factors for performance.
	- The number of goals generated changes fundamentally the search space.
- In the main body, it should say that BestFS with a trained policy, and how it's implemented. (See my comment on contribution 2)

# Minor questions or comments

- Page 5: > Rubik’s Cube is a celebrated 3D puzzle with over 4.3 × 10 18possible configurations.
	- Please cite Korf's paper
- Page 9. Verifier: precision and recall
	- I’d say the verifier increase accuracy only in a small budget in INT.
- Page 6: "Shaded areas indicate 95% confidence intervals." How many trials?
- Page 15, Sect "Computational budget analysis"
	- says: Tables 3, 2 and 4 present the number of calls to each component per 1000 episodes.
	- Is this testing or during training?
	- Are those 1000 instances the same for all the algorithms?
	- Is that the total number for 1000 episodes? The common practice in deterministic search is to report numbers per episode.
- page 17:
	- > Rubik’s Cube. To construct a single successful trajectory we performed 20 random permutations on an initially solved Rubik’s Cube and took the reverse of this sequence. Using this procedure we collected 10 7trajectories"
	- are there symmetric movements? Are these optimals path? See comment by Clemens et al.
- Page 18:
	- > "ρ BFSworks in the following way. First, it uses a trained policy network to generate actions to investigate. Specifically, for INT we use beam search to generate high probability actions (it is necessary as for INT we represent actions as sequences, following Czechowski et al. (2021)). Then, it uses these actions to get a state that follows a given action (note that all our environments are deterministic). Finally, we treat returned states as our new subgoals, which are easily found in one step by the low-level policy."
	 - It's not clear what happens in other domains.
- page 15: please say this in the main body, not just in the appendix:
	- > used for sampling predictions from the subgoal generators – the only component that outputs a set of predictions.


**Summary Of The Paper:**

The paper presents an algorithm –AdaSubS– that can be trained and configured to solve deterministic search problems. The key idea is to consider multiple lookaheads that would be explored with lower-level policy. The ablation study confirms the effectiveness of the idea in challenging domains.


**Summary Of The Review:**

The ideas are novel, the components of the algorithm are well-studied to understand their role, and the benchmarks are appropriate.

I recommend acceptance, but I might lower my scores depending on the answer to my questions.

---

> ### Author Response · Authors · 2022-11-19
> **Answer to the Reviewer Kkdv**
>
> We are grateful to the Reviewer for the extensive review and multiple valid points! The comments are highly relevant and provided us with valuable suggestions to improve our paper. The suggested competition also sounds fun, we will certainly look into that!
>
> Regarding the specific questions:
> - Czechowski et al. (2021) should be mentioned as a starting point.
>
> We added a comment on that in the second paragraph of the introduction.
>
> - Scholarship on search and planning
>
> We thank the reviewer for pointing out the relevant references. We made the appropriate modifications in Section 2 (related work) and Section 4.4 (developing adaptive search methods).
>
> - No limitations are discussed (see my first question below as an instance)
>
> We discussed the limitations in Section 5. We extended this part by discussing memory constraints (related to the next item) and adversarial configuration (inspired by the stress scenario described by the Reviewer). We also extended Appendix C with a discussion of memory usage.
>
> - Does the algorithm keep a list of close nodes? Do you cache the result of evaluations in Alg 1 or Alg 2?
>
> The algorithm visits every node at most once, which is guaranteed by keeping a “list” of already seen subgoals (data structure seen in Alg 1). Consequently, each network is evaluated at most once for each node and does not need caching. However, Alg 4 can benefit from caching outputs of GET_PATH in Alg 1. This optimization is present in the actual source code for the algorithm. We put the corresponding comment in Alg 4. We keep Alg 1 unchanged to maintain its clarity and reduce boilerplate.
>
> - How would AdaSubS behave in the following case?
>
> It is true that the performance of AdaSubS depends on the quality of its components.  In the suggested scenario, the chance of randomly generating the correct k-subgoal decreases geometrically with k. Consequently, the performance for higher k might be impaired. This, in turn, suggests using small values k. This corresponds to the case where each state is complex, and retraction to a shorter subgoal is advised.
>
> Having said that, the way the training protocol is structured inclines the generators to avoid dead ends since they never appear as subgoal targets during training. Finally, notice that in a scenario where not all components are bad, the good ones can compensate for the weak ones. If a value function is good, it should guide the search in the right direction, despite weak subgoal generators. Conversely, if the generators work well, most subgoals should offer progress, even for a weak value function.
>
> - Would you comment on the following paper published in IJCAI 2021?
>
> We thank the Reviewer for bringing Allen et al. (2021) to our attention. It has several similarities to our work, and we now include it in the reference section. Having said that, the main differences are as follows. First, Allen et al. (2021) generate a set of macro-actions (a series of actions) independent of the node the search is visiting during inference. This procedure does not involve training and utilizes a hand-crafted notion of distance to the goal supplied along with the environment (the so-called goal-count heuristic). In our work, we train neural network subgoal state generator models. This is a data-driven approach, and it was found by Czechowski et al. (2021) that for some environments, generating states perform much better than generating actions. The generators are state-dependent (i.e., subgoals depend on the current node visited by the planner), and the overall procedure is adaptive. Second, the ability to create useful macro-actions in the manner proposed by Allen et al. (2021) depends on the size of the action space. On the contrary, our approach is data-driven, so its effectiveness is restricted by the size of the training dataset (which is not an issue if access to data is unrestricted). For the same reason, we do not need access to the goal-count heuristic, which may or may not be available, and may or may not be informative. These problems can be visualized in the example of INT, where the action space is effectively infinite (which makes constructing the first layer in the search for macro-actions virtually impossible), and it is unclear what would be a useful goal-count heuristic. We have included a summary of this discussion in Section 2.
>
> - Why does the evaluation support the 2nd main contribution?
>
> We agree with the Reviewer’s suggestion, and we made the suggested change.

---

> > ### Author Response · Authors · 2022-11-19
> > **Answer to the Reviewer Kkdv continued**
> >
> > - How were the hyperparameters set? Are the hyper-parameters of BestFS set in the best possible way? BestFS is not explained well enough.
> >
> > We extended Appendix F with a description of tuning the hyperparameters (see Appendix F.1): the ranges, the required number of experiments, and the estimated computational cost. We estimate that the whole process requires ten training runs on  GPU and 40 CPU evaluations to make a hyperparameter sweep and determine a good set of values for each environment.
> >
> > We put considerable effort into tuning the BestFS hyperparameters. Most importantly, the main component of BestFS is the value network, which is shared by all the tested algorithms. To further clarify the BestFS baseline, we extended its description in Appendix E with a pseudocode.
> >
> > We think setting hyperparameters for AdaSubS is comparable to typical deep learning hyperparameters tuning pipeline. Additionally, we laid out a blueprint for choosing k for subgoal generators and calibrating verifier thresholds in Appendix F.1.
> >
> >
> > We also thank the Reviewer for pointing out minor issues; these have been reviewed and corrected. In particular:
> >
> > - Other changes
> >
> > We now mention the number of subgoals generators and the values of k in Section 3. We extended the description of BestFS in Appendix E, referenced in the main body. Although we would like to discuss its details in the main body as well, we can’t because of the tight space constraints.
> >
> > - Minor questions or comments
> >
> > Ad 95% confidence intervals, we have clarified that a fixed set of 1000 problems was used for each domain (Figure 2 caption). Regarding Rubik’s Cube data, we clarified that the movements are not symmetric themselves, but we reverse them when collecting the paths (Appendix D). This usually leads to suboptimal trajectories. As to ρ BFS, we clarified the description in Appendix E. We also stressed in the main text that the generator is the only component that outputs a set of predictions (Section 3).
> >
> > - I’d say the verifier increases accuracy only in a small budget in INT.
> >
> > The advantage of using the verifier varies, depending on the budget and environment. Perhaps unsurprisingly, it is most visible with small budgets and longer subgoal distances. For the INT environment, we use rather short subgoals (1, 2, and 3). On average, even if modest, the verifier brings improvements (see Table 1 and Tables 10-12 in Appendix for complete numerical results). For this reason, we argue that it is a useful mechanism to keep in the toolbox.
> >
> > - Can actual V values be the same for two different nodes?
> >
> > Surprisingly, the answer is yes. We observed two different Sokoban boards (for the same game) with exactly the same value, up to numerical precision. However, it happens rarely – less than one time for 1k problem solutions.

---

> > > ### Comment · Reviewer_Kkdv · 2022-11-22
> > > **Thank you**
> > >
> > > Thank you for your comments and improvements. I saw the changes in the paper, and I think some of the other answers should lead to changes there.
> > >
> > > In particular, the comment about classical planning and multiple heuristics is still falling short. Too short. One of the key points of the proposed algorithm is that, so it's fair to discuss it more at length in the paper.
> > > This point is as important as "hierarchical planning utilizing different temporal distances" in related work, that takes a whole paragraph.
> > > If you need more space, I suggest reducing the number of references in the 2nd paragraph of related work, and add a section in the appendix with further references.
> > >
> > > I'm not trying to weaken the paper but to make it stronger by drawing deeper connections with the literature. Perhaps this observation would end up being connected with ensembles and multi-task learning.
> > >
> > > After reading other reviews, I think the comment on memory should be seen as completeness. Having a list of visited nodes is necessary for being able to cover the whole state space.

---

> > > > ### Author Response · Authors · 2022-11-23
> > > > **Answer to the Reviewer Kkdv**
> > > >
> > > > We are grateful for the Reviewer's extensive effort to strengthen our paper. We followed the Reviewer's advice and extended the related work section. These are important topics that place our work in a broader context. Below we present the revised part of the section. We can also provide a link to an updated anonymized PDF (however, it is not clear to us whether it does not breach the ICLR’s rules).
> > > >
> > > > We also agree that there is a strong connection between completeness and memory constraints. We merged those two points in the limitations section.
> > > >
> > > > If the Reviewer has any further comments, please do not hesitate to let us know.
> > > >
> > > > *Our approach relates to established search algorithms (Cormen et al. (2009); Russell & Norvig (2010)), such as Best First Search or A\*. Adaptivity techniques in the classical setup are discussed in Fickert (2022). Domain-independent PDDL-based planners McDermott et al. (1998) do not use training and attempt to solve problems in a zero-shot manner. Thus, they indicate a lower bound on performance. On the other hand, there are domain-specific methods (Korf (1997); Büchner et al. (2022); Muppasani et al. (2022)). Due to their focus, they indicate an upper bound.*
> > > >
> > > > *AdaSubS relates to multi-queue methods (see Richter & Westphal (2010); Helmert (2006)), which alternate between multiple heuristics. Some of our planners, e.g., IterativeMixing or Longest-first, can be viewed through the lens of this approach in the sense that we could keep the priority queues for each generator separate (but with the same heuristic being a value function) and have an alternation mechanism between them. The key difference lies in the expansion phase: we expand subgoals instead of children, and only the generator associated with the currently selected queue is used. For search engines using multi-queues, see Fast downward https://www.fast-downward.org/; LAPKT https://github.com/LAPKT-dev/LAPKT-public. For PDDL generators, see https://github.com/AI-Planning/pddl-generators.*
> > > >
> > > > *Different instances of Sokoban, Rubik’s Cube, and INT can be viewed as tasks with varying degrees of difficulty. Consequently, AdaSubS benefits from sharing data between these instances in a manner typical to multitask learning Caruana (1997).
> > > > In particular, we use a goal-conditioned policy which is trained similarly as in Andrychowicz et al. (2018); Kaelbling (1993). Additionally, the out-of-distribution generalization of AdaSubS hints at strong meta-learning capabilities of the method Yu et al. (2019); Duan et al. (2016); Wang et al. (2016), Hassel et al. (2018).*
> > > >
> > > > References:
> > > > - [Cormen et al. (2009)] Introduction to algorithms, Cormen et al, 2009
> > > > - [Russell & Norvig (2010)] Artificial intelligence: A modern approach. Russell & Norvig, 2009
> > > > - [Fickert (2022)] Adaptive search techniques in ai planning and heuristic search, Fickert, 2022
> > > > - [McDermott et al. (1998)] PDDL - The Planning Domain Definition Language, McDermott et al, 1998
> > > > - [Korf (1997)] Finding optimal solutions to rubik’s cube using pattern databases, Korf, 1997
> > > > - [Büchner et al. (2022)] A comparison of abstraction heuristics for rubik’s cube. Büchner et al, 2022
> > > > - [Muppasani et al. (2022)] Solving the rubik’s cube with a pddl planner. Muppasani et al, 2022
> > > > - [Richter & Westphal (2010)] The lama planner: Guiding cost-based anytime planning with landmarks. Richter & Westphal, 2010
> > > > - [Helmert (2006)] The fast downward planning system. Helmert, 2006
> > > > - [Caruana (1997)] Multitask learning, R. Caruana, 1997
> > > > - [Andrychowicz et al. (2018)] Hindsight Experience Replay, Andrychowicz et al, 2018
> > > > - [Kaelbling (1993)] Learning to achieve goals, Kaelbling, 1993
> > > > - [Yu et al. (2019)] Meta-World: A Benchmark and Evaluation for Multi-Task and Meta Reinforcement Learning, Yu et al, 2019
> > > > - [Duan et al. (2016)] RL^2: Fast Reinforcement Learning via Slow Reinforcement Learning, Duan et al, 2016
> > > > - [Wang et al. (2016)] Learning to reinforcement learn, Wang et al, 2016
> > > > - [Hassel et al. (2018)] Multi-task Deep Reinforcement Learning with PopArt, Hassel et al, 2018

---

> > > > > ### Comment · Reviewer_Kkdv · 2022-11-23
> > > > > **Good addition**
> > > > >
> > > > > I understand the text in cursive would be added to the paper.
> > > > > That’s very good.

---

> > > > > > ### Author Response · Authors · 2022-11-24
> > > > > > **Thank you.**
> > > > > >
> > > > > > Yes, the cursive fragment has already been added to our internal version of the paper. We do not post it here (as a link) as, again, in a strict interpretation, this would violate the ICLR rules.
> > > > > >
> > > > > > And again, thank you for helping us.

---

### Public Comment · ~Kyo_Takano1 · 2022-11-12
**A note on DeepCubeA & an additional reference**

Dear authors and reviewers,


I would like to make a few comments about references, although they may not be very helpful since the authors do not seem to compare their results with existing ones.

1. DeepCubeA (Agostinelli et al., 2019) [1] is a method that has been validated on Sokoban as well as the Rubik's Cube. The authors has not included this information, like Czechowski, et al. (2021) [2] did not.
2. On solving the Rubik's Cube, we proposed a deep learning method with state-of-the-art results (Takano, 2021) [3], which also combined neural networks and BestFS (more specifically, beam search).

Please consider including/citing these. If possible, with experimental comparisons of solution optimality and efficiency.

---
References

[1] Forest Agostinelli, Stephen McAleer, Alexander Shmakov, and Pierre Baldi. Solving the Rubik’s cube with deep reinforcement learning and search. Nature Machine Intelligence, 1(8):356–363, 2019.\
[2] Konrad Czechowski, Tomasz Odrzygózdz, Marek Zbysinski, Michał Zawalski, Krzysztof Olejnik, Yuhuai Wu, Łukasz Kucinski, and Piotr Miłos. Subgoal search for complex reasoning tasks. Advances in Neural Information Processing Systems, 34:624–638, 2021.\
[3] Kyo Takano. Self-Supervision is All You Need for Solving Rubik's Cube. arXiv:2106.03157, 2021.

---

> ### Author Response · Authors · 2022-11-19
> **Answer**
>
> Thank you for taking an interest in our work and for your remarks! We feel that it is always instructive to see what happens with new baselines. Having said that,  we will not be able to make a proper set of experiments (which includes coding, debugging, tuning, and analysis) of a new method within the timeframe of the rebuttal.
>
> We already cite [1], and thank you for bringing [3] to our attention. Is there a published version of this paper?

---

### Author Response · Authors · 2022-11-19
**Overall Reviewer Response**

We want to thank all Reviewers for taking the time to review our work and providing relevant and vital feedback. We made multiple improvements to the paper suggested by the Reviewers (color-coded in orange). We are grateful for the appreciation of novelty (KKdv, EUhR, 1TzK, 1BT4), quality of experiments (KKdv, EUhR, 1TzK, 1BT4), state-of-the-art on INT (EUhR, 1BT4), comprehensive analysis and ablations (Kkdv, 1BT4), and we are pleased that the paper was found to be well-written (KKdv, EUhR, 1BT4). Please see our answers posted below as separate comments for the Reviewer's individual concerns.

---

### Decision · Program_Chairs · 2023-01-20

**Decision:**

Accept: notable-top-5%

**Justification For Why Not Higher Score:**

I have given the highest score.

**Justification For Why Not Lower Score:**

The only reason for giving this a lower score might be that it is not of sufficient interest to the general ICLR community to justify an oral.  However, the idea is novel and the empirical results seem strong on meaningful (not toy) applications.

**Metareview: Summary, Strengths And Weaknesses:**

This paper presents an algorithm for adaptive subgoal horizons.  This appears to be a novel technique with significant potential.  Inititial concerns about scholarship have been addressed by the authors with updates to the paper.  There was considerable discussion between the reviewers and the authors and the paper has been generally improved as a result.

The main weakness is perhaps that the results are not surprising given that the value adaptivity in planning is well established.  But important ideas often look easy or trivial in retrospect.

**Note From Pc:**

if the above contains the word "oral" or "spotlight" please see: "oral" presentation means -> notable-top-5% and "spotlight" means -> notable-top-25%. As stated in our emails, we are disassociating presentation type from AC recommendations